# DIAGNOSING AND IMPROVING DIFFUSION MODELS BY ESTIMATING THE OPTIMAL LOSS VALUE

**Yixian Xu**[1]*, **Shengjie Luo**[1]*, **Liwei Wang**[1], **Di He**[1]†, **Chang Liu**[2]†
[1] State Key Laboratory of General Artificial Intelligence, Peking University, Beijing, China
[2] Zhongguancun Academy, Beijing, China

## ABSTRACT

Diffusion models have achieved remarkable success in generative modeling. Despite more stable training, the loss of diffusion models is not indicative of absolute data-fitting quality, since its optimal value is typically not zero but unknown, leading to the confusion between large optimal loss and insufficient model capacity. In this work, we advocate the need to estimate the optimal loss value for diagnosing and improving diffusion models. We first derive the optimal loss in closed form under a unified formulation of diffusion models, and develop effective estimators for it, including a stochastic variant scalable to large datasets with proper control of variance and bias. With this tool, we unlock the inherent metric for diagnosing training quality of mainstream diffusion model variants, and develop a more performant training schedule based on the optimal loss. Moreover, using models with 120M to 1.5B parameters, we find that the power law is better demonstrated after subtracting the optimal loss from the actual training loss, suggesting a more principled setting for investigating the scaling law for diffusion models.

## 1 INTRODUCTION

Diffusion-based generative models (Sohl-Dickstein et al., 2015; Ho et al., 2020; Song et al., 2021b) have shown unprecedented capability in modeling high-dimensional distribution and have become the dominant choice in various domains. The attractive potential has incentivized advances in multiple dimensions, such as prediction targets (Kingma et al., 2021; Salimans and Ho, 2022; Lipman et al., 2023), diffusion process design (Karras et al., 2022; Liu et al., 2023), and training schedule design (Nichol and Dhariwal, 2021; Kingma and Gao, 2023; Esser et al., 2024).

The success is largely benefited from the more stable training process. Nevertheless, the diffusion loss only reflects the *relative* data-fitting quality for monitoring training process or comparing models under the same setting, while remains obscure for measuring the *absolute* fit to the training data. It is due to that the optimal loss of diffusion model, *i.e.*, the lowest possible loss value that can be attained by any model, is actually not zero but *unknown* beforehand. This introduces a series of inconveniences. After the training converges, one still does not know whether the model is already close to oracle, or the remaining loss can be further reduced by tuning the model. Practitioners have to rely on generating samples to evaluate diffusion models, which requires significant computational cost, and sampler configurations introduce distracting factors. The unknown optimal loss also makes it obscured to analyze and compare learning quality at different diffusion steps, impeding a principled design of training schedule. Moreover, as the actual loss value is not fully determined by model capacity but also the unknown optimal loss as the base value, it poses a question on using the actual loss value alone for monitoring the scaling law of diffusion models.

In this work, we highlight the importance of estimating the optimal loss value, and develop effective estimation methods applicable to large datasets. Using this tool, we unlock new observations of data-fitting quality of diffusion models under various formulation variants, and demonstrate how the optimal loss estimate leads to more principled analysis and performant designs. Specifically,

- We reveal the indefiniteness of the optimal loss from its expression, then develop estimators for the optimal loss based on the expression. For large datasets, we design a scalable estimator based on dataset sub-sampling, with a delicate design to properly balance variance and bias.

---

*Equal contribution.
†Correspondence to: Di He <dihe@pku.edu.cn>, Chang Liu <liuchang@bza.edu.cn>.

- Using the estimator, we reveal the patterns of the optimal loss across diffusion steps on diverse datasets, and by comparison with the losses of mainstream diffusion models under a unified formulation, we find the characteristics of different diffusion formulation variants, and identify the diffusion-step region where the model still underfits compared to the optimal loss.

- From the analysis, we designed a principled training schedule for diffusion models, based on the gap between the actual loss and the optimal loss. Our training schedule improves the FID by 2%-14% (for EDM (Karras et al., 2022) / FM (Lipman et al., 2023)) on CIFAR-10, 7%-25% (for EDM / FM) on ImageNet-64, and 9% (for LightningDiT (Yao et al., 2025)) on ImageNet-256.

- We challenge the conventional formulation to study neural scaling law for diffusion models. We propose using the loss gap as the measure for data-fitting quality. Using state-of-the-art diffusion models (Karras et al., 2024) in various sizes from 120M to 1.5B on both ImageNet-64 and ImageNet-512, we find that our modification leads to better satisfaction of the power law.

We would mention that estimating the optimal loss is not meant to achieve it, which may render overfitting, but to introduce a metric for measuring the absolute fitness to a dataset (Appx. A). We hope this work could provide a profound understanding on diffusion model training, and ignite more principled analyses and improvements for diffusion models.

## 1.1 RELATED WORK

**Optimal loss and solution of diffusion model.** A related work by Bao et al. (2022b;a) derived the optimal ELBO loss under discrete Gaussian reverse process, and used it to determine the optimal reverse Gaussian (co)variances and optimize the discrete diffusion steps. Gu et al. (2023) further studied the memorization behavior of diffusion models. In contrast, we consider general cases and develop effective training-free estimators for the optimal loss value, and emphasize its principled role with important real examples in monitoring and diagnosing model training, designing training schedules, and studying the scaling law. There are also some other works that made efforts to estimate the optimal solution (Xu et al., 2023) using importance sampling. Although more scalable methods are proposed using fast KNN search (Niedoba et al., 2024), their viability for estimating the optimal loss on large datasets remains unverified, as the optimal loss requires estimating two nested expectations (see Appx. G.2).

**Training design of diffusion model.** Due to the stochastic nature, intensive research efforts are paid to investigate diffusion model training in multiple directions such as noise schedules and loss weight. Karras et al. (2022) presented a design space that clearly separates design choices, enabling targeted explorations on training configurations. Kingma and Gao (2023) analyzed different diffusion objectives in a unified way and connect them via ELBO. Esser et al. (2024) conduct large-scale experiments to compare different training configurations and motivate scalable design choices for billion-scale models. Most works require large-scale compute for trial and error, due to the lack of a principled guideline for training schedule design based on the absolute data-fitting quality.

**Scaling law study for diffusion model.** Model scaling behaviors are of great interest in deep learning literature. In particular, the remarkable success of Large Language Models has been largely credited to the establishment of scaling laws (Kaplan et al., 2020; Henighan et al., 2020; Hoffmann et al., 2022), which help to predict the performance of models as they scale in parameters and data. There also exist works that empirically investigate the scaling behavior of diffusion models (Peebles and Xie, 2023; Li et al., 2024; Mei et al., 2025; Esser et al., 2024), and make attempts to explicitly formulate scaling laws for diffusion transformers (Liang et al., 2024). However, training loss values are typically used as the metric in these works, which are not corrected by the optimal loss to reflect the true optimization gap, leading to biased analysis for scaling behaviors of diffusion models.

## 2 FORMULATION OF DIFFUSION MODEL

Diffusion models perform generative modeling by leveraging a step-by-step transformation from an arbitrary data distribution $p_{\text{data}}$ to a Gaussian distribution. Sampling and density evaluation for the data distribution can be done by reversing this transformation process step by step from the Gaussian. In general, the transformation of distribution is constructed by:

$$\mathbf{x}_t = \alpha_t \mathbf{x}_0 + \sigma_t \boldsymbol{\epsilon}, \quad t \in [0, T], \tag{1}$$

where $\mathbf{x}_0 \sim p_{\text{data}}$ is taken as a data sample, $\boldsymbol{\epsilon} \sim p(\boldsymbol{\epsilon}) := \mathcal{N}(\mathbf{0}, \mathbf{I})$ is a Gaussian noise sample, and $\mathbf{x}_t$ is the constructed random variable that defines the intermediate distribution $p_t$. The coefficients

$\alpha_t$ and $\sigma_t$ satisfy $\alpha_0 = 1$, $\sigma_0 = 0$, and $\alpha_T \ll \sigma_T$, so that $p_0 = p_{\text{data}}$ and $p_T = \mathcal{N}(\mathbf{0}, \sigma_T^2 \mathbf{I})$ yield the desired distributions. Eq. (1) gives $p(\mathbf{x}_t \mid \mathbf{x}_0) = \mathcal{N}(\mathbf{x}_t \mid \alpha_t \mathbf{x}_0, \sigma_t^2 \mathbf{I})$, which can be achieved by a diffusion process expressed in the stochastic differential equation $d\mathbf{x}_t = a_t \mathbf{x}_t \, dt + g_t \, d\mathbf{w}_t$ starting from $\mathbf{x}_0 \sim p_0$, where $a_t := (\log \alpha_t)'$, $g_t := \sigma_t \sqrt{(\log \sigma_t^2/\alpha_t^2)'}$, and $\mathbf{w}_t$ denotes the Wiener process. The blessing of the diffusion-process formulation is that the reverse process can be given explicitly (Anderson, 1982):

$$d\mathbf{x}_s = -a_{T-s}\mathbf{x}_s \, ds + g_{T-s}^2 \nabla \log p_{T-s}(\mathbf{x}_s) \, ds + g_{T-s} \, d\mathbf{w}_s$$

from $\mathbf{x}_{s=0} \sim p_T$, where $s := T - t$ denotes the reverse time. Alternatively, the deterministic process given by the ordinary differential equation:

$$d\mathbf{x}_s = -a_{T-s}\mathbf{x}_s \, ds + \frac{1}{2}g_{T-s}^2 \nabla \log p_{T-s}(\mathbf{x}_s) \, ds$$

produces the same distribution $p_{T-s}$ at each reverse diffusion step $s$. The only obstacle to simulating the reverse process for generation is the unknown term $\nabla \log p_t(\mathbf{x}_t)$ called the score function. Noting that $p_t$ is produced by perturbing data samples with Gaussian noise, diffusion models employ a neural network model $\mathbf{s}_{\boldsymbol{\theta}}(\mathbf{x}_t, t)$ to learn the score function using the denoising score matching loss (Vincent, 2011; Song et al., 2021b): $J_t^{(\mathbf{s})}(\boldsymbol{\theta}) :=$

$$\mathbb{E}_{p_0(\mathbf{x}_0)p(\mathbf{x}_t|\mathbf{x}_0)}\|\mathbf{s}_{\boldsymbol{\theta}}(\mathbf{x}_t, t) - \nabla_{\mathbf{x}_t}\log p(\mathbf{x}_t|\mathbf{x}_0)\|^2 \overset{\text{Eq. (1)}}{=} \mathbb{E}_{p_0(\mathbf{x}_0)p(\boldsymbol{\epsilon})}\|\mathbf{s}_{\boldsymbol{\theta}}(\alpha_t \mathbf{x}_0 + \sigma_t \boldsymbol{\epsilon}, t) + \boldsymbol{\epsilon}/\sigma_t\|^2. \quad (2)$$

To cover the whole diffusion process, loss weight $w_t^{(\mathbf{s})}$ and noise schedule $p(t)$ are introduced to optimize over all diffusion steps using the total loss $J(\boldsymbol{\theta}) := \mathbb{E}_{p(t)}w_t^{(\mathbf{s})}J_t^{(\mathbf{s})}(\boldsymbol{\theta})$.

**Alternative prediction targets.** Besides the above *score prediction* target, diffusion models also adopt other prediction targets. Eq. (2) motivates the *noise prediction* ($\boldsymbol{\epsilon}$-prediction) target (Ho et al., 2020) $\boldsymbol{\epsilon}_{\boldsymbol{\theta}}(\mathbf{x}_t, t) := -\sigma_t \mathbf{s}_{\boldsymbol{\theta}}(\mathbf{x}_t, t)$, which turns the loss into:

$$J_t^{(\boldsymbol{\epsilon})}(\boldsymbol{\theta}) := \mathbb{E}_{p_0(\mathbf{x}_0)}\mathbb{E}_{p(\boldsymbol{\epsilon})}\|\boldsymbol{\epsilon}_{\boldsymbol{\theta}}(\alpha_t \mathbf{x}_0 + \sigma_t \boldsymbol{\epsilon}, t) - \boldsymbol{\epsilon}\|^2 = \sigma_t^2 J_t^{(\mathbf{s})}(\boldsymbol{\theta}), \quad (3)$$

It poses a friendly, bounded-scale learning target, and avoids the artifact at $t = 0$ of $J_t^{(\mathbf{s})}(\boldsymbol{\theta})$. If formally solving $\mathbf{x}_0$ from Eq. (1) and let $\mathbf{x}_{0\boldsymbol{\theta}}(\mathbf{x}_t, t) := \frac{\mathbf{x}_t - \sigma_t \boldsymbol{\epsilon}_{\boldsymbol{\theta}}(\mathbf{x}_t, t)}{\alpha_t}$, then we get the loss:

$$J_t^{(\mathbf{x}_0)}(\boldsymbol{\theta}) := \mathbb{E}_{p_0(\mathbf{x}_0)}\mathbb{E}_{p(\boldsymbol{\epsilon})}\|\mathbf{x}_{0\boldsymbol{\theta}}(\alpha_t \mathbf{x}_0 + \sigma_t \boldsymbol{\epsilon}, t) - \mathbf{x}_0\|^2 = (\sigma_t^4/\alpha_t^2)J_t^{(\mathbf{s})}(\boldsymbol{\theta}). \quad (4)$$

It holds the semantics of *clean-data prediction* ($\mathbf{x}_0$-prediction) (Kingma et al., 2021; Karras et al., 2022), and can be viewed as denoising auto-encoders (Vincent et al., 2008; Alain and Bengio, 2014) with multiple noise scales. From the equivalent deterministic process, one can also derive the *vector-field prediction* (v-prediction) target $\mathbf{v}_{\boldsymbol{\theta}}(\mathbf{x}_t, t) := a_t \mathbf{x}_t - \frac{1}{2}g_t^2 \mathbf{s}_{\boldsymbol{\theta}}(\mathbf{x}_t, t)$ with loss function

$$J_t^{(\mathbf{v})}(\boldsymbol{\theta}) := \mathbb{E}_{p_0(\mathbf{x}_0)}\mathbb{E}_{p(\boldsymbol{\epsilon})}\|\mathbf{v}_{\boldsymbol{\theta}}(\alpha_t \mathbf{x}_0 + \sigma_t \boldsymbol{\epsilon}, t) - (\alpha_t'\mathbf{x}_0 + \sigma_t'\boldsymbol{\epsilon})\|^2 = (g_t^4/4)J_t^{(\mathbf{s})}(\boldsymbol{\theta}). \quad (5)$$

It coincides with the velocity prediction (Salimans and Ho, 2022) and the flow matching formulation (Lipman et al., 2023; Liu et al., 2023) ($\alpha_t'\mathbf{x}_0 + \sigma_t'\boldsymbol{\epsilon}$ is the conditional vector field). See Appx. B for details.

## 3 ESTIMATING THE OPTIMAL LOSS VALUE FOR DIFFUSION MODELS

The diffusion loss in various forms (Eqs. 2-5) allows effective and stable learning of intractable targets that would otherwise require diffusion simulation or posterior estimation. Nevertheless, as we will show from the expression of the optimal solution and loss (Sec. 3.1), the optimal loss value is typically non-zero but unknown, obscuring the diagnosis and design of diffusion training. We then develop practical estimators of the optimal loss value, starting from a standard one (Sec. 3.2) to stochastic but scalable estimators applicable to large datasets (Sec. 3.3). Using these tools, we investigate mainstream diffusion models against the optimal loss with a few new observations (Sec. 3.4).

### 3.1 OPTIMAL SOLUTION AND LOSS VALUE OF DIFFUSION MODELS

Despite the intuition, the names of the prediction targets of diffusion model introduced in Sec. 2 might be misleading. Taking the clean-data prediction formulation as an example, it is informationally impossible to predict the exact clean data from its noised version (Daras et al., 2023). From the appearance of the loss functions (Eq. (2-5)), the actual learning targets of the models are *conditional*

*expectations* (De Bortoli et al., 2021; Bao et al., 2022b;a):

$$\mathbf{s}_{\boldsymbol{\theta}}^{\star}(\mathbf{x}_t, t) = \mathbb{E}_{p(\mathbf{x}_0|\mathbf{x}_t)}[\nabla_{\mathbf{x}_t} \log p(\mathbf{x}_t \mid \mathbf{x}_0)], \qquad \boldsymbol{\epsilon}_{\boldsymbol{\theta}}^{\star}(\mathbf{x}_t, t) = \mathbb{E}_{p(\boldsymbol{\epsilon}|\mathbf{x}_t)}[\boldsymbol{\epsilon}],$$

$$\mathbf{x}_{0\boldsymbol{\theta}}^{\star}(\mathbf{x}_t, t) = \mathbb{E}_{p(\mathbf{x}_0|\mathbf{x}_t)}[\mathbf{x}_0], \qquad \mathbf{v}_{\boldsymbol{\theta}}^{\star}(\mathbf{x}_t, t) = \mathbb{E}_{p(\mathbf{x}_0, \boldsymbol{\epsilon}|\mathbf{x}_t)}[\alpha_t' \mathbf{x}_0 + \sigma_t' \boldsymbol{\epsilon}],$$

where the conditional distributions are induced from $p(\mathbf{x}_0, \mathbf{x}_t, \boldsymbol{\epsilon}) := p_0(\mathbf{x}_0)p(\boldsymbol{\epsilon})\delta_{\alpha_t \mathbf{x}_0 + \sigma_t \boldsymbol{\epsilon}}(\mathbf{x}_t)$. For completeness, we detail the derivation in Appx. C.

Looking back into the loss functions, the model learns the conditional expectations over random samples from the joint distribution. Hence even at optimality, the loss still holds a conditional variance value. Noting that the joint distribution hence the conditional variance depends on the data distribution, it would be more direct to write down the optimal loss value in the clean-data prediction formulation, which we formally present below:

**Theorem 1.** *The optimal loss value for the clean-data prediction target defined in Eq. (4) is:*

$$J_t^{(\mathbf{x}_0)\star} = \underbrace{\mathbb{E}_{p_0(\mathbf{x}_0)}\|\mathbf{x}_0\|^2}_{=:A} - \underbrace{\mathbb{E}_{p(\mathbf{x}_t)}\|\mathbb{E}_{p(\mathbf{x}_0|\mathbf{x}_t)}[\mathbf{x}_0]\|^2}_{=:B_t}, \quad J^{\star} = \mathbb{E}_{p(t)}w_t^{(\mathbf{x}_0)} J_t^{(\mathbf{x}_0)\star}. \tag{6}$$

See Appx. E.1 for proof. For other prediction targets, the optimal loss value can be calculated based on their relations in Eqs. (3, 4, 5) . The expression is derived from $J_t^{(\mathbf{x}_0)\star} = \mathbb{E}_{p_t(\mathbf{x}_t)} \operatorname{tr} \operatorname{Cov}_{p(\mathbf{x}_0|\mathbf{x}_t)}[\mathbf{x}_0]$, which is indeed an averaged conditional variance, hence takes a positive value unless at $t = 0$ or when $p_0(\mathbf{x}_0)$ concentrates only on a single point. For a small $t$, $\mathbf{x}_t$ is close to the $\mathbf{x}_0$ that produces it via noising and is unlikely to be produced by other $\mathbf{x}_0$, so $p(\mathbf{x}_0 \mid \mathbf{x}_t)$ has a low variance and $J_t^{(\mathbf{x}_0)\star}$ approaches zero as $t \to 0$. $J_t^{(\mathbf{x}_0)\star}$ increases with $t$, until for a sufficiently large $t$, $\mathbf{x}_t$ becomes dominated by the noise (see Eq. (1)) hence has a diminishing correlation with $\mathbf{x}_0$, so $p(\mathbf{x}_0 \mid \mathbf{x}_t) \approx p_{\text{data}}(\mathbf{x}_0)$ and $J_t^{(\mathbf{x}_0)\star} \approx \operatorname{tr} \operatorname{Cov}_{p_{\text{data}}(\mathbf{x}_0)}[\mathbf{x}_0]$ approaches the data variance. Note that this optimal loss only depends on dataset and diffusion settings, but not on model architectures and parameterization.

## 3.2 EMPIRICAL ESTIMATOR FOR THE OPTIMAL LOSS VALUE

To estimate the optimal loss value using Eq. (6) on a dataset $\{\mathbf{x}_0^{(n)}\}_{n \in [N]}$, where $[N] := \{1, \cdots, N\}$, the first term $A := \mathbb{E}_{p_0(\mathbf{x}_0)}\|\mathbf{x}_0\|^2$ can be directly estimated through one pass:

$$\hat{A} = \frac{1}{N} \sum_{n \in [N]} \|\mathbf{x}_0^{(n)}\|^2. \tag{7}$$

However, the second term $B_t := \mathbb{E}_{p(\mathbf{x}_t)}\|\mathbb{E}_{p(\mathbf{x}_0|\mathbf{x}_t)}[\mathbf{x}_0]\|^2$ requires estimating two nested expectations that cannot be reduced. The inner expectation is taken under the posterior distribution $p(\mathbf{x}_0 \mid \mathbf{x}_t)$ which cannot be sampled directly. By expanding the distribution using tractable ones (Bayes rule), the term can be reformulated as: $\mathbb{E}_{p(\mathbf{x}_0|\mathbf{x}_t)}[\mathbf{x}_0] =$

$$\frac{\mathbb{E}_{p(\mathbf{x}_0)}[\mathbf{x}_0 p(\mathbf{x}_t \mid \mathbf{x}_0)]}{\mathbb{E}_{p(\mathbf{x}_0)}[p(\mathbf{x}_t \mid \mathbf{x}_0)]} = \frac{\mathbb{E}_{p(\mathbf{x}_0)}[\mathbf{x}_0 K_t(\mathbf{x}_t, \mathbf{x}_0)]}{\mathbb{E}_{p(\mathbf{x}_0)}[K_t(\mathbf{x}_t, \mathbf{x}_0)]}, \text{ where } K_t(\mathbf{x}_t, \mathbf{x}_0) := \exp\left\{-\frac{\|\mathbf{x}_t - \alpha_t \mathbf{x}_0\|^2}{2\sigma_t^2}\right\}, \tag{8}$$

according to Eq. (1). Both the numerator and denominator in the expression can then be estimated on the dataset. The outer expectation can be estimated by averaging over a set of independent and identically distributed (IID) samples $\{\mathbf{x}_t^{(m)}\}_{m \in [M]}$ following Eq. (1), where each sample is produced by an independently (*i.e.*, with replacement) randomly selected data sample $\mathbf{x}_0$ and a randomly drawn noise sample $\boldsymbol{\epsilon} \sim \mathcal{N}(\mathbf{0}, \mathbf{I})$. The estimator for the second term is then:

$$\hat{B}_t = \frac{1}{M} \sum_{m \in [M]} \left\|\frac{\sum_{n \in [N]} \mathbf{x}_0^{(n)} K_t(\mathbf{x}_t^{(m)}, \mathbf{x}_0^{(n)})}{\sum_{n' \in [N]} K_t(\mathbf{x}_t^{(m)}, \mathbf{x}_0^{(n')})}\right\|^2. \tag{9}$$

The outer expectation can be conducted sequentially until the estimation converges. This typically takes $M$ up to two to three times of $N$. See Appx. G.1 for details.

## 3.3 SCALABLE ESTIMATORS FOR LARGE DATASETS

Although asymptotically unbiased (Appx. F), the $\hat{B}$ estimator in Eq. (9) incurs a quadratic complexity in dataset size $N$, which is unaffordably costly for large datasets which are ubiquitous in modern machine learning tasks. For a scalable estimator, dataset sub-sampling is an effective strategy to reduce computational complexity. Instead of using independent random subsets to estimate the

numerator and denominator separately, we adopt the self-normalized importance sampling (SNIS) estimator (Robert et al., 1999; Kroese and Rubinstein, 2012) (see Appx. F for background):

$$\hat{B}_t^{\text{SNIS}} := \frac{1}{M} \sum_{m \in [M]} \left\| \frac{\sum_{l \in [L]} \mathbf{x}_0^{(l)} K_t(\mathbf{x}_t^{(m)}, \mathbf{x}_0^{(l)})}{\sum_{l' \in [L]} K_t(\mathbf{x}_t^{(m)}, \mathbf{x}_0^{(l')})} \right\|^2 .$$

It uses the same randomly selected (with replacement) subset $\{\mathbf{x}_0^{(l)}\}_{l \in [L]}$, where $L \ll N$, for both the numerator and denominator, which leads to more stable estimates. One can repeat drawing the random data subset $\{\mathbf{x}_0^{(l)}\}_{l \in [L]}$ and calculate the estimate until convergence.

A specialty for estimating the diffusion optimal loss is that, for a given $\mathbf{x}_t^{(m)}$ sample, when $\sigma_t$ is small, the weight term $K_t(\mathbf{x}_t^{(m)}, \mathbf{x}_0^{(l)})$ is dominated by the $\mathbf{x}_0$ sample closest to $\mathbf{x}_t^{(m)}/\alpha_t$ (see Eq. (8)), which could be missed in the randomly selected subset $\{\mathbf{x}_0^{(l)}\}_{l \in [L]}$, thus incurring a large variance. Fortunately, we know that by construction (Eq. (1)), each $\mathbf{x}_t^{(m)}$ sample is produced from a data sample $\mathbf{x}_0^{(n_m)}$ and a noise sample $\boldsymbol{\epsilon}^{(m)}$ using $\mathbf{x}_t^{(m)} = \alpha_t \mathbf{x}_0^{(n_m)} + \sigma_t \boldsymbol{\epsilon}^{(m)}$, and when $\sigma_t$ is small, $\alpha_t$ is also close to 1 (Sec. 2), indicating that $\mathbf{x}_0^{(n_m)}$ is likely the most dominant $\mathbf{x}_0$ sample and should be included in the subset $\{\mathbf{x}_0^{(l)}\}_{l \in [L]}$. This can be simply implemented by constructing the $\{\mathbf{x}_t^{(\tilde{m})}\}_{\tilde{m} \in [M]}$ samples by independently (*i.e.*, with replacement) drawing a sample $\mathbf{x}_0^{(l_{\tilde{m}})}$ from the subset $\{\mathbf{x}_0^{(l)}\}_{l \in [L]}$ and setting $\mathbf{x}_t^{(\tilde{m})} = \alpha_t \mathbf{x}_0^{(l_{\tilde{m}})} + \sigma_t \boldsymbol{\epsilon}^{(\tilde{m})}$ with $\boldsymbol{\epsilon}^{(\tilde{m})} \sim \mathcal{N}(\mathbf{0}, \mathbf{I})$. We call it the Diffusion Optimal Loss (DOL) estimator:

$$\hat{B}_t^{\text{DOL}} := \frac{1}{M} \sum_{\tilde{m} \in [M]} \left\| \frac{\sum_{l \in [L]} \mathbf{x}_0^{(l)} K_t(\mathbf{x}_t^{(\tilde{m})}, \mathbf{x}_0^{(l)})}{\sum_{l' \in [L]} K_t(\mathbf{x}_t^{(\tilde{m})}, \mathbf{x}_0^{(l')})} \right\|^2 . \tag{10}$$

Nevertheless, this introduces an artificial correlation between $\mathbf{x}_t$ and $\mathbf{x}_0$ samples: it becomes more probable to calculate $K_t$ for $(\mathbf{x}_t, \mathbf{x}_0)$ pairs where $\mathbf{x}_t$ is constructed from $\mathbf{x}_0$. Such pairs have larger $K_t$ values, hence over-estimating $B_t$ and under-estimating the optimal loss $J_t^{(\mathbf{x}_0)^\star}$. We introduce a simple correction by down-weighting such pairs with a coefficient $C$, and call it the corrected DOL (cDOL) estimator:

$$\hat{B}_t^{\text{cDOL}} := \frac{1}{M} \sum_{\tilde{m} \in [M]} \left\| \frac{\sum_{l \in [L], l \neq l_{\tilde{m}}} \mathbf{x}_0^{(l)} K_t(\mathbf{x}_t^{(\tilde{m})}, \mathbf{x}_0^{(l)}) + \frac{1}{C} \mathbf{x}_0^{(l_{\tilde{m}})} K_t(\mathbf{x}_t^{(\tilde{m})}, \mathbf{x}_0^{(l_{\tilde{m}})})}{\sum_{l' \in [L], l' \neq l_{\tilde{m}}} K_t(\mathbf{x}_t^{(\tilde{m})}, \mathbf{x}_0^{(l')}) + \frac{1}{C} K_t(\mathbf{x}_t^{(\tilde{m})}, \mathbf{x}_0^{(l_{\tilde{m}})})} \right\|^2 , \tag{11}$$

where $l_{\tilde{m}}$ indexes the sample in $\{\mathbf{x}_0^{(l)}\}_{l \in [L]}$ that is used to construct $\mathbf{x}_t^{(\tilde{m})}$. To formalize the effectiveness, we provide the following theoretical result on the cDOL estimator:

**Theorem 2.** *The $\hat{B}_t^{\text{cDOL}}$ estimator with subset size $L$ has the same expectation as the $\hat{B}_t^{\text{SNIS}}$ estimator with subset size $L - 1$ when $M \to \infty, C \to \infty$, hence is a consistent estimator.*

See Appx. E.2 for proof. Note that the first terms in the numerator and denominator are unbiased, but the second terms introduce biases due to the artificial correlation between $\mathbf{x}_t$ and $\mathbf{x}_0$ samples. The DOL estimator in Eq. (10) amounts to using $C = 1$, which suffers from the biases. The bias can be reduced using $C > 1$ in the cDOL estimator. On the other hand, the second terms become the dominant components at small $t$ for estimating the numerator and denominator, respectively. Always including them using a finite $C$ hence reduces estimation variance. The complete process of the cDOL estimator is concluded in Alg. 1 in Appx. G.1.

### 3.4 ESTIMATION RESULTS OF OPTIMAL LOSS VALUES

We now provide empirical results of diffusion optimal loss estimates on popular datasets. We first compare the scalable estimators on CIFAR-10 (Krizhevsky et al., 2009) and FFHQ-64 (Karras, 2019) (Fig. 1(a,b)), whose relatively small sizes allow the full-dataset estimate by Eqs. (7, 9) , providing a reference for the scalable estimators. With the best scalable estimator identified (from Fig. 2), we apply it to the much larger ImageNet-64 (Krizhevsky et al., 2012) dataset, and analyze the optimal loss pattern (Fig. 1(c)).

As different prediction targets (Sec. 2) and diffusion processes (Sec. 4.1 below) can be converted to each other, we choose the clean-data prediction target and variance exploding (VE) process ($\alpha_t \equiv 1$) (Song and Ermon, 2019; Song et al., 2021b) to present the diffusion optimal loss. We plot the optimal loss for each diffusion step, which is marked by $\log \sigma$ to decouple the arbitrariness in the time schedule $\sigma_t$ (as advocated by (Karras et al., 2022); the same $\sigma$ indicates the same distribution at that step of diffusion). All the scalable estimators repeat data subset sampling until the estimate converges. See Appx. G.1 for settings and discussions on efficiency.

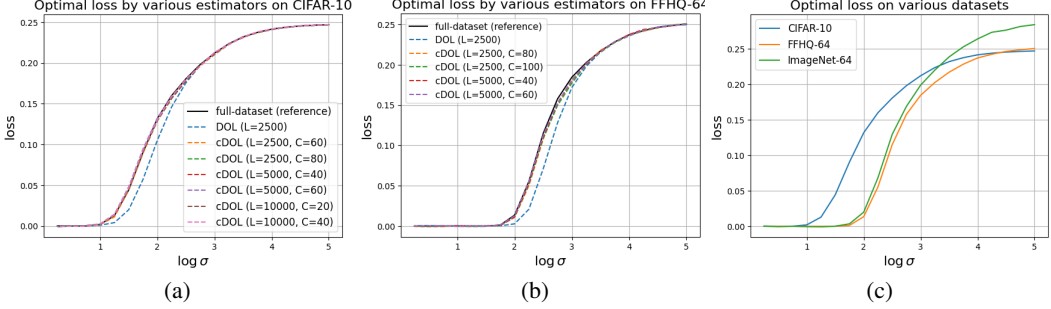

Figure 1: Estimation results of optimal loss value. **(a,b)** Noise-scale-wise optimal loss estimates by the DOL (Eqs. 7, 10) and the corrected DOL (cDOL) (Eqs. 7, 11) estimators, with the full-dataset estimate (Eqs. 7, 9) as reference, on the **(a)** CIFAR-10 and **(b)** FFHQ-64 datasets. **(c)** Noise-scale-wise optimal loss on various datasets in different scales. Figures are plotted for the $\mathbf{x}_0$-prediction loss under the VE process.

**Comparison among the scalable estimators.** From Fig. 1(a,b), we can see that the DOL estimator indeed underestimates the optimal loss as we pointed out, especially at intermediate diffusion steps. The cDOL estimator can effectively mitigate the bias, and stays very close to the reference under diverse choices of $C$. The insensitivity of the cDOL estimator w.r.t $C$ can be understood as that, for small $t$ (equivalently, $\sigma$), both the numerator and denominator are dominated by the $C$-corrected terms, in which $C$ cancels out, and for large $t$, the $K_t(\mathbf{x}_t^{(\tilde{m})}; \mathbf{x}_0^{(l\tilde{m})})$ term is in the same scale as other terms hence is overwhelmed when compared with the summation.

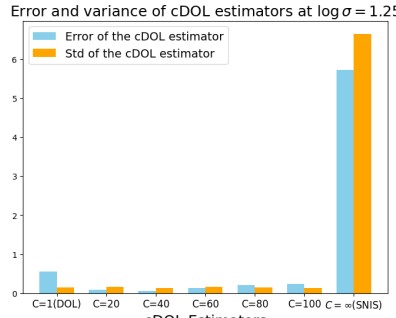

Figure 2: Error and variance of cDOL estimates using various $C$ values (including DOL and SNIS as extreme cases) for the optimal loss at $\log \sigma = 1.25$ on CIFAR-10.

To better analyze the behavior of the estimators, we zoom in on their estimation error and standard deviation. Fig. 2 presents the results at an intermediate $\log \sigma$ where the estimation is more challenging. The result confirms that the variance increases with $C$. Particularly, at $C = \infty$ which corresponds to the SNIS estimator (Thm. 2), it is hard to sample the dominating cases for the estimate, leading to a large variance, and a significantly large estimation error. At the $C = 1$ end which corresponds to the DOL estimator, although the variance is smaller, its bias still leads it to a large estimation error. The cDOL estimator with $C$ in between achieves consistently low estimation error. Empirically, a preferred $C$ is around $4N/L$. The subset size $L$ can be taken to fully utilize memory. We also compare our cDOL estimators with prior estimators for optimal *solution*; see Appx. G.2.

**The pattern of optimal loss.** From Fig. 1(c), we observe that the optimal loss $J_t^{(\mathbf{x}_0)^\star}$ increases monotonically with the noise scale $\sigma$ on all the three datasets. The optimal loss is close to zero only when the noise scale $\sigma$ is less than a *critical point* $\sigma^\star$, in which situation the noisy samples stay so close to their corresponding clean sources that they are unlikely to intersect with each other, hence preserve the information of the clean samples, allowing the model to perform a nearly perfect denoising. We can see that the critical point $\sigma^\star$ depends on the dataset. CIFAR-10 achieves the minimal $\sigma^\star$, since it has the lowest image resolution (32×32), *i.e.*, the lowest data-space dimension, where the data samples appear less sparse hence easier to overlap after isotropic noise perturbation. Both FFHQ-64 and ImageNet-64 have 64×64 resolution, but ImageNet-64 is larger, hence data samples are easier to overlap, leading to a smaller $\sigma^\star$.

Beyond the critical point, the optimal loss takes off quickly. The positive value indicates the intrinsic difficulty of the denoising task, where even an oracle denoiser would be confused. The increase trend converges for sufficiently large noise scale $\sigma$, which meets our analysis under Thm. 1 that $J_t^{(\mathbf{x}_0)^\star}$ converges to the data variance. As ImageNet-64 contains more diverse samples (images from more classes), it has a larger data variance, hence converges to a higher value than the other two.

Table 1: Viewing mainstream diffusion models under the same formulation as $\mathbf{x}_0$-prediction under the VE process, following Eq. (12). Each diffusion model is labeled by "diffusion process"-"prediction target" ("common name").

| Formulations | $c_\sigma^{\text{skip}}$ | $c_\sigma^{\text{out}}$ | $c_\sigma^{\text{in}}$ | $c_\sigma^{\text{noise}}$ | $w_\sigma$ | $p(\sigma)$ |
|---|---|---|---|---|---|---|
| VP-$\boldsymbol{\epsilon}$ (DDPM) (Ho et al., 2020) | $1$ | $-\sigma$ | $\frac{1}{\sqrt{1+\sigma^2}}$ | $999\,t(\sigma)$ | $\frac{1}{\sigma^2}$ | $t \sim \mathcal{U}(10^{-5}, 1),$ $\sigma = \sqrt{e^{\beta_{\min}t + \frac{1}{2}(\beta_{\max}-\beta_{\min})t^2} - 1}$ |
| VE-$\mathbf{F}$ (EDM) (Karras et al., 2022) | $\frac{\sigma_{\text{data}}^2}{\sigma^2+\sigma_{\text{data}}^2}$ | $\frac{\sigma_{\text{data}}\sigma}{\sqrt{\sigma^2+\sigma_{\text{data}}^2}}$ | $\frac{1}{\sqrt{\sigma^2+\sigma_{\text{data}}^2}}$ | $\frac{1}{4}\log\sigma$ | $\frac{\sigma^2+\sigma_{\text{data}}^2}{\sigma^2\sigma_{\text{data}}^2}$ | $\log\sigma \sim \mathcal{N}(P_{\text{mean}}, P_{\text{std}}^2)$ |
| VE-$\boldsymbol{\epsilon}$ (NCSN) (Song et al., 2021a) | $1$ | $\sigma$ | $1$ | $\log\frac{\sigma}{2}$ | $\frac{1}{\sigma^2}$ | $\log\sigma \sim \mathcal{U}(\log\sigma_{\min}, \log\sigma_{\max})$ |
| FM-$\mathbf{v}$ (FM) (Lipman et al., 2023) | $\frac{1}{1+\sigma}$ | $-\frac{\sigma}{1+\sigma}$ | $\frac{1}{1+\sigma}$ | $\frac{\sigma}{1+\sigma}$ | $(\frac{1+\sigma}{\sigma})^2$ | $t \sim \mathcal{U}(0,1), \sigma = \frac{t}{1-t}$ |
| FM-$\mathbf{v}$ (SD3) (Esser et al., 2024) | $\frac{1}{1+\sigma}$ | $-\frac{\sigma}{1+\sigma}$ | $\frac{1}{1+\sigma}$ | $\frac{\sigma}{1+\sigma}$ | $(\frac{1+\sigma}{\sigma})^2$ | $\log\sigma \sim \mathcal{N}(0,1)$ |

# 4 ANALYZING AND IMPROVING DIFFUSION TRAINING SCHEDULE WITH OPTIMAL LOSS

From the training losses in Sec. 2, the degree of freedom for the training strategy is the *noise schedule* $p(t)$ and the *loss weight* $w_t$, collectively called the training schedule. In the literature, extensive works (Ho et al., 2020; Song et al., 2021b; Karras et al., 2022; Kingma and Gao, 2023; Esser et al., 2024) have designed training schedules for various prediction targets and diffusion processes individually, based on the analysis on the loss scale over diffusion steps. Here, we argue that analyzing the *gap* between the actual loss and the optimal loss would be a more principled approach, since it is the gap but not the loss itself that reflects the data-fitting insufficiency and the potential for improvement. Under this view, we first analyze and compare the loss gap of mainstream diffusion works on the same ground (Sec. 4.1), identifying new patterns that are related to inference (*i.e.*, generation) performance. We then develop a new training schedule based on the observation (Sec. 4.2).

## 4.1 ANALYZING TRAINING SCHEDULES THROUGH OPTIMAL LOSS

Existing training schedules are developed under different diffusion processes and prediction targets. For a unified comparison on the same ground, we start with the equivalence among the formulations and convert them to the same formulation. As explained in Sec. 3.4, we use the noise scale $\sigma$ in place of $t$ to mark the diffusion step to decouple the choice of time schedule $\sigma_t$.

**Equivalent conversion among diffusion formulations.** Sec. 2 has shown the equivalence and conversion among prediction targets. We note that different diffusion processes in the form of Eq. (1) can also be equivalently converted to each other. Particularly, the variance preserving (VP) process ($\alpha_\sigma = \sqrt{1-\sigma^2}$) (Sohl-Dickstein et al., 2015; Ho et al., 2020) and the flow matching (FM) process ($\alpha_\sigma = 1-\sigma$) (Lipman et al., 2023; Liu et al., 2023) can be converted to the variance exploding (VE) process ($\alpha_\sigma \equiv 1$) (Song and Ermon, 2019; Song et al., 2021b) by $\mathbf{x}_\sigma^{\text{VE}} := \frac{\mathbf{x}_\sigma}{\alpha_\sigma}$, since $\mathbf{x}_\sigma^{\text{VE}} = \mathbf{x}_0 + \frac{\sigma}{\alpha_\sigma}\boldsymbol{\epsilon}$ by Eq. (1), and $\mathbf{x}_0 = \mathbf{x}_0^{\text{VE}}$. The correspondence of diffusion step is given by $\sigma^{\text{VE}} = \frac{\sigma}{\alpha_\sigma}$. With this fact, various diffusion models can be viewed as different parameterizations of the $\mathbf{x}_0$-prediction under the VE process (Karras et al., 2022), where the parameterization is formulated by:

$$\mathbf{x}_{0\boldsymbol{\theta}}(\mathbf{x}, \sigma) = c_\sigma^{\text{skip}}\mathbf{x} + c_\sigma^{\text{out}}\mathbf{F}_{\boldsymbol{\theta}}(c_\sigma^{\text{in}}\mathbf{x}, c_\sigma^{\text{noise}}), \tag{12}$$

where $\mathbf{x}_{0\boldsymbol{\theta}}$, $\mathbf{x}$, and $\sigma$ are the $\mathbf{x}_0$-prediction model, the diffusion variable, and the noise scale under the VE process, and $\mathbf{F}_{\boldsymbol{\theta}}(\cdot, \cdot)$ represents the bare neural network used for the original prediction target and diffusion process. The precondition coefficients $c_\sigma^{\cdot}$ are responsible for the conversion. Their instances for reproducing mainstream diffusion models are listed in Table 1, where the converted $w_\sigma$ and $p(\sigma)$ from the original works are also listed. See Appx. D for details. For EDM (Karras et al., 2022), the precondition coefficients are not derived from a conversion but directly set to satisfy the input-output unit variance principle. This leads to a new prediction target we call the $\mathbf{F}$-prediction.

**Comparison between optimal loss gaps.** Using the above equivalent convention, we convert the actual training loss of diffusion models for various prediction targets under various diffusion processes to the $\mathbf{x}_0$-prediction loss under the VE process, which serves as a unified metric on the same ground. We conduct the comparison on the CIFAR-10 dataset, and compare the gap between their actual loss and the optimal loss, which has been estimated and presented in Fig. 1(a; the full-dataset curve). Results shown in Fig. 3(a) reveal some new observations. The optimal loss gap across different diffusion steps is not even: most of the representative diffusion models leave a large loss gap

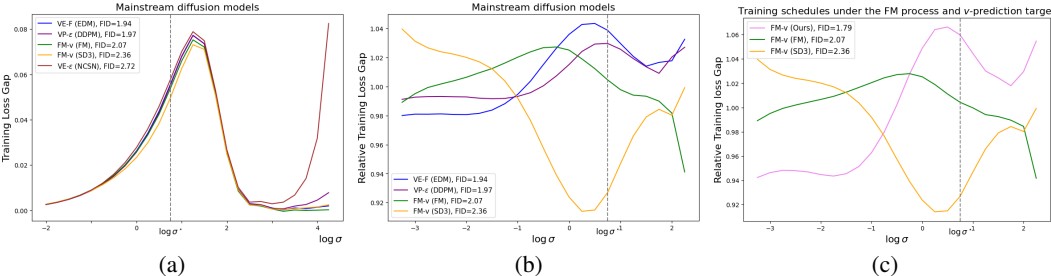

Figure 3: Training loss gap across diffusion noise scales under various diffusion model training settings on CIFAR-10. FID as a metric for inference performance is marked for each training setting in the legend. **(a)** Training loss gap of mainstream diffusion models. **(b)** Relative training loss gap for a clearer comparison among mainstream diffusion models. The relative values at each noise scale are taken by dividing the (absolute) training loss gap values under the four settings by their average value (*i.e.*, noise-scale-wise normalized). "VE-$\boldsymbol{\epsilon}$" is omitted for its salient difference. **(c)** Relative training loss gap for comparing existing training schedules and our schedule (Sec. 4.2) under the FM process with the $\mathbf{v}$-prediction target. Curves under different diffusion model settings are plotted together by viewing them as parameterizations of the $\mathbf{x}_0$-prediction under the variance exploding (VE) process (Eq. (12); Table 1). See Appx. G.3 for detailed settings, results using different samplers and under precision/recall and memorization metrics.

in the intermediate diffusion steps around $\log \sigma \in [-2, 2]$, indicating room to improve. In addition, $\boldsymbol{\epsilon}$-prediction models incur a large error for large $\sigma$, revealing a difficulty in learning such models.

**Loss gap vs. inference performance.** We now use the gap to the optimal loss as the fundamental data-fitting measure to analyze which region is more critical for inference (*i.e.*, generation) performance, measured in Fréchet Inception Distance (FID) (Heusel et al., 2017) also marked in Fig. 3(a). All diffusion models use the same deterministic ODE sampler with NFE = 35 following Karras et al. (2022). We can see that an erroneous fit at large noise scales of $\boldsymbol{\epsilon}$-prediction models leads to a deficiency in generation quality, *e.g.*, NCSN vs EDM in Fig. 3(a). Among methods with a good fit for large $\sigma$, the intermediate noise scale region $\log \sigma \in [-2.0, 2.0]$ becomes more relevant to inference performance. We then zoom into the intermediate region and compare these methods using the normalized training loss gap, taken by dividing the value on each curve by the average value over the four curves, as shown in Fig. 3(b) ("VE-$\boldsymbol{\epsilon}$" is omitted due to its significant deviation from other curves). Counter-intuitively, around the critical point $\sigma^\star$ defined as the largest $\sigma$ whose optimal loss $J_\sigma^\star$ becomes positive, although the gap is around the largest over noise scales, the loss gap turns out to *negatively* correlate with FID. The correlation becomes *positive* only in the region further left to $\sigma^\star$. This reveals that there is a trade-off in learning a diffusion model at different noise scales, and sacrificing the fit in the region around $\sigma^\star$ for a better fit in the noise scale interval further left to $\sigma^\star$ leads to a better inference performance.

## 4.2 PRINCIPLED DESIGN OF TRAINING SCHEDULE

From the observation above, the training schedule plays a crucial role in optimizing diffusion models, due to the trade-off over different noise scales. We hence design a principled training schedule based on conclusions from the representative optimal loss estimates.

**The loss weight.** $w_\sigma$ calibrates the error resolution across different noise scales. For this, the optimal loss $J_\sigma^\star$ provides a perfect reference scale for the loss at each diffusion step $\sigma$, so $w_\sigma = a/J_\sigma^\star$ with a scale factor $a$ is a natural choice to align the loss at various $\sigma$ to the same scale. Although it downweighs the loss for large noise scales, the above observation suggests that the model can still achieve a good fit if using $\mathbf{v}$-prediction and $\mathbf{F}$-prediction (Table 1). For smaller noise scales, a cutoff $w^\star$ is needed to avoid divergence, which stops the increase of $w_\sigma$ before $\sigma$ runs smaller than the critical point $\sigma^\star$. As observed in Sec. 4.1, the interval where the loss gap has a positive correlation to inference performance is on the left of (vs. around) $\sigma^\star$. We hence introduce an additional weight function $f(\sigma) = \mathcal{N}(\log \sigma; \mu, \varsigma^2)$, whose parameters $\mu$ and $\varsigma$ are chosen to put the major weight over the left interval. The resulting loss weight is finally given by:

$$w_\sigma = a \min\left\{1/J_\sigma^\star, w^\star\right\} + f(\sigma) \mathbb{I}_{\sigma < \sigma^\star}. \tag{13}$$

The loss weight for other prediction targets can be derived following conversions in Eqs. (3, 4, 5) .

Table 3: Comparison between existing training schedules and our schedule on ImageNet-256.

| Method | Generation w/o CFG | | | | Generation w/ CFG | | | |
|---|---|---|---|---|---|---|---|---|
| | FID($\downarrow$) | IS($\uparrow$) | Pre.($\uparrow$) | Rec.($\uparrow$) | FID($\downarrow$) | IS($\uparrow$) | Pre.($\uparrow$) | Rec.($\uparrow$) |
| *Pixel-space Diffusion Models* | | | | | | | | |
| ADM (Dhariwal and Nichol, 2021) | 10.94 | – | 0.69 | 0.63 | 3.94 | 215.9 | 0.83 | 0.53 |
| RIN (Jabri et al., 2022) | 3.42 | 182.0 | – | – | – | – | – | |
| SimpleDiffusion (Hoogeboom et al., 2023) | 2.77 | 211.8 | – | – | 2.12 | 256.3 | – | – |
| VDM++ (Kingma and Gao, 2023) | 2.40 | **225.3** | – | – | 2.12 | 267.7 | – | – |
| SiD2 (Hoogeboom et al., 2024) | – | – | – | – | 1.38 | – | – | – |
| *Latent Diffusion Models* | | | | | | | | |
| MaskDiT (Zheng et al., 2023) | 5.69 | 177.9 | 0.74 | 0.60 | 2.28 | 276.6 | 0.80 | 0.61 |
| DiT (Peebles and Xie, 2023) | 9.62 | 121.5 | 0.67 | 0.67 | 2.27 | 278.2 | **0.83** | 0.57 |
| SiT (Ma et al., 2024) | 8.61 | 131.7 | 0.68 | 0.67 | 2.06 | 270.3 | 0.82 | 0.59 |
| FasterDiT (Yao et al., 2024) | 7.91 | 131.3 | 0.67 | **0.69** | 2.03 | 264.0 | 0.81 | 0.60 |
| MDT (Gao et al., 2023a) | 6.23 | 143.0 | 0.71 | 0.65 | 1.79 | 283.0 | 0.81 | 0.61 |
| MDTv2 (Gao et al., 2023b) | – | – | – | – | 1.58 | **314.7** | 0.79 | 0.65 |
| REPA (Yu et al., 2024) | 5.90 | – | – | – | 1.42 | 305.7 | 0.80 | 0.65 |
| LightningDiT (Yao et al., 2025) | 2.17 | 205.6 | **0.77** | 0.65 | 1.35 | 295.3 | 0.79 | 0.65 |
| + reproduction | 2.29 | 206.2 | 0.76 | 0.66 | 1.42 | 292.9 | 0.79 | 0.65 |
| **+ Our schedule** | **2.08** | 220.8 | **0.77** | 0.66 | **1.30** | 301.3 | 0.79 | **0.66** |

**The noise schedule.** $p(\sigma)$ allocates the optimization frequency to each noise level. A desired $p(\sigma)$ should favor noise steps on which the optimization task has not yet been done well, which can be measured by the difference from $w_\sigma J_\sigma(\theta)$ to $w_\sigma J_\sigma^\star$. This provides a principled measure for optimization insufficiency, which leads to an adaptive noise schedule: $p(\sigma) \propto w_\sigma (J_\sigma(\theta) - J_\sigma^\star)$. See Fig. G.4.1 for an example of the resulting noise schedule in practical training, which indeed allocates more steps in the positively correlated interval.

Note that applying the loss weight and noise schedule in training a model only requires estimating the optimal loss $J_\sigma^\star$ on the training dataset, which does not require training a model beforehand.

**CIFAR-10 & ImageNet-64 Results.** We evaluate the designed training schedule in training two advanced diffusion models EDM (Karras et al., 2022) and Flow Matching (FM) (Lipman et al., 2023) on the CIFAR-10 and ImageNet-64 datasets (details in Appx. G.4). As shown in Table 2, our training schedule significantly improves inference (*i.e.*, generation) performance upon the original for both the EDM and FM settings on both datasets, demonstrating the value of new insights from our analysis using the optimal loss. For

Table 2: Generation FID ($\downarrow$) by existing training schedules and ours on CIFAR-10 and ImageNet-64.

| | CIFAR-10 | | ImageNet-64 |
|---|---|---|---|
| | Conditional | Unconditional | Conditional |
| StyleGAN (Karras, 2019) | 2.42 | 2.92 | - |
| ScoreSDE (deep) (Song et al., 2021b) | - | 2.20 | - |
| ImprovedDDPM(Nichol and Dhariwal, 2021) | - | 2.94 | 3.54 |
| 2-Rectified Flow (Liu et al., 2023) | - | 4.85 | 2.92 |
| VDM (Kingma et al., 2021) | - | 2.49 | 3.40 |
| EDM (Karras et al., 2022) | 1.79 | 1.98 | 2.44 |
| + EDM2 (Karras et al., 2024) schedule | 1.94 | 2.09 | - |
| **+ Our schedule** | **1.75** | **1.94** | **2.25** |
| FM (Lipman et al., 2023) | - | 6.35 | 14.45 |
| + EDM sampler | 2.07 | 2.24 | 3.06 |
| **+ Our schedule** | **1.77** | **2.03** | **2.29** |

a closer look at how our schedule works, in Fig. 3(c) we plot the relative training loss gap across noise scales using our schedule, and compare it with the schedule of the original (Lipman et al., 2023) and of StableDiffusion 3 (SD3) (Esser et al., 2024). We find that our schedule indeed further decreases the loss in the interval with positive correlation to performance, aligning with the insight from Sec. 4.1.

**ImageNet-256 Results.** Finally, we evaluate our training schedule on the ImageNet-256 dataset and compare the results with existing approaches. We use VA-VAE (Yao et al., 2025) as the tokenizer and employ a modified LightningDiT (Yao et al., 2025) architecture enhanced with QK-Normalization (Dehghani et al., 2023) to improve training stability (details in Appx. G.4). As shown in Table 3, our training schedule further improves inference performance over the original LightningDiT training schedule. Human evaluation results in Table G.4.1 also suggest the advantages.

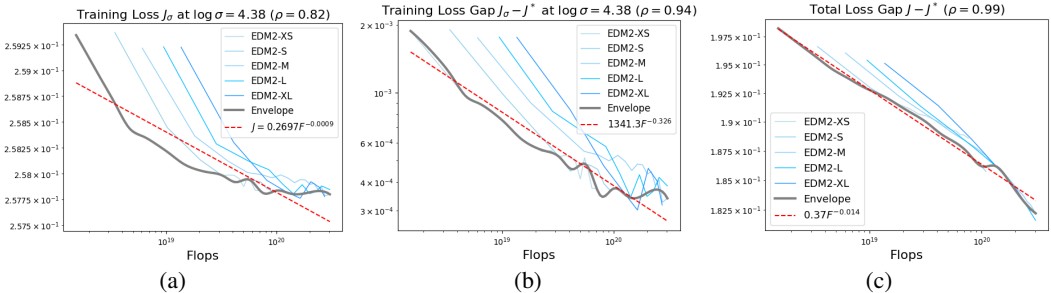

Figure 4: Scaling law study using optimal loss on ImageNet-64. Training curves and their envelopes for a range of model sizes showing **(a)** the actual absolute loss value and **(b)** the gap between the actual and the optimal loss at noise scale $\log \sigma = 4.38$. **(c)** Training curves and their envelope showing the gap for the total loss (averaged over noise scales according to the training schedule). Correlation coefficients $\rho$ for the envelopes are marked.

## 5 PRINCIPLED SCALING LAW STUDY FOR DIFFUSION MODELS

Neural scaling law (Kaplan et al., 2020) has been the driving motivation for pursuing large models, which shows the consistent improvement of model performance with computational cost. The conventional version takes the form of a power law (Kaplan et al., 2020; Henighan et al., 2020; Hoffmann et al., 2022): $J(F) = \beta F^{\alpha}$, where $F$ denotes floating point operations (FLOPs) measuring training budget, $J(F)$ denotes the minimal training loss attained by models in various sizes (the envelope in Fig. 4), and $\alpha < 0$ and $\beta > 0$ are power law parameters. The specialty of a scaling law study for diffusion model is that, as the optimal loss sets a non-zero lower bound of the training loss, not all the loss value in $J(F)$ can be reduced with the increase of $F$, questioning the form which converges to zero as $F \to \infty$. Instead, the following modified power law is assumed:

$$J(F) - J^{\star} = \beta F^{\alpha}, \tag{14}$$

where $J^{\star}$ denotes the optimal loss value. It can be rephrased as that $\log(J(F) - J^{\star})$ is linear in $\log F$, so we can verify it through the linear correlation coefficient $\rho$. We conduct experiments using current state-of-the-art diffusion models EDM2 (Karras et al., 2024) with parameter size ranging from 120M to 1.5B.

We first compare the training curves at a large noise scale, for which Fig. 4(a) and (b) assume the original and the modified (Eq. (14)) scaling laws, respectively. We can observe that in the modified version, the envelope is indeed closer to a line, and the improved correlation coefficient $\rho = 0.94$ (vs. $0.82$) validates this quantitatively. For the total loss, we use the optimized adaptive loss weight by EDM2 (Karras et al., 2024). The result is shown in Fig. 4(c), which achieves $\rho = 0.9917$, and the fitted scaling law is given by: $J(F) = 0.3675 \, F^{-0.014} + 0.015$. Appx. G.5 provides more results on ImageNet-512. We hope this approach could lead to more profound future studies in the scaling law for diffusion models.

## 6 CONCLUSION

In this work, we emphasize the central importance of optimal loss estimation to make the training loss value meaningful for diagnosing and improving diffusion model training. To the best of our knowledge, we are the first to notice and work on this issue. We develop analytical expressions and practical estimators for diffusion optimal loss, particularly a scalable estimator applicable to large datasets and has proper variance and bias control. With this tool, we revisit the training behavior of mainstream diffusion models, and propose an optimal-loss-based approach for a principled design of training schedules, which indeed improves performance in practice. Furthermore, we investigate the scaling behavior w.r.t model size, in which accounting for the optimal loss value as an offset makes the scaling curves better fulfill the power law.

Although the optimal loss estimation is not directly meant for analyzing inference performance, it introduces a serious metric for data-fitting quality, which paves the way for future generalization studies. We believe our new approaches and insights could motivate future research on analyzing and improving diffusion models, and advance the progress of generative-model research.

ACKNOWLEDGMENTS

This work is supported by Zhongguancun Academy (Grant No. C20250506). DH is supported by National Science Foundation of China (NSFC62376007), National Science Foundation of China (under Key Project No. 92570203), Beijing Natural Science Foundation (Z250001) and Beijing Major Science and Technology Project under Contract no. Z251100008425004.

## 7 ETHICS STATEMENT

This work adheres to the ICLR Code of Ethics. Our study does not involve human subjects, personal data, or sensitive demographic information. All experiments are conducted on publicly available benchmark datasets, which are widely used in the machine learning community. No new data collection or human/animal experimentation was performed.

## 8 REPRODUCIBILITY STATEMENT

To facilitate the reproducibility of our research, we provide comprehensive details throughout the paper and its supplementary materials. We begin by establishing the necessary backgrounds in Sec. 2. For all theoretical claims in the main text, we offer detailed derivations in Appx. E. All our experiments are thoroughly documented; the datasets, training procedures, and settings are carefully described in Appx. G. Upon acceptance of this paper, we commit to making our full codebase and all model checkpoints publicly available to ensure that the community can fully reproduce our results.

## 9 THE USE OF LARGE LANGUAGE MODELS (LLMS)

In the preparation of this manuscript, LLMs were employed as a writing assistant to refine the language and improve the grammar. Following this process, all textual content was meticulously reviewed, revised, and validated by the authors, who assume full responsibility for the final work presented.

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

APPENDIX

The supplementary material is organized as follows. We first remark the relation between estimating optimal loss and achieving generalization in Appx. A. In Appx. B, we review alternative formulations of diffusion models. In Appx. C, we provide a detailed derivation of the optimal solution for diffusion models in various formulations. In Appx. D, we give the detailed derivation of the conversion shown in Table 1. In Appx. E, we give the proofs of all theorems. In Appx. F, we give a brief introduction to the important sampling methods. In Appx. G, we give the details of all our experiments.

## A    DISCUSSIONS ON OPTIMAL LOSS ESTIMATION AND GENERALIZATION IN DIFFUSION MODELS

We would like to emphasize that the motivation of estimating the optimal loss is *not* to enforce the model to achieve the optimal loss on a dataset, in which case the model is overfitted to the training data samples, but to fulfill the more fundamental need of measuring the absolute fitness to a dataset, as a monitor to the training process, and also the evaluation of fitness to a held-out test set. As an analogy, monitoring the supervision loss (*e.g.*, the mean squared error (MSE) between model prediction and data labels) does not meant to optimize it to zero on the training set, but to evaluate the fitness to data. Particularly, one would use the loss to monitor training process, and the use the supervision loss on a test set to evaluate the performance, which is essentially to evaluate the fitness to the test set. The specialty with diffusion loss is that its optimal value is unknown beforehand, so the loss value does not readily reflect the absolute fitness to a dataset. Estimating the diffusion optimal loss can hence provide a reference to interpret a real training loss value. This even enables evaluating the generalization of a diffusion model using the loss value on a test set compared to the optimal value on the dataset. This evaluation does not require the costly and tricky generation process as is typically adopted.

For the design of the training schedule, we would like to mention that the design is not intended to reduce the loss gap for every time step. As is shown in Fig. 3, there is a trade-off of loss optimization at different time steps, where the loss gap in some time-step intervals is positively correlated to inference (*i.e.*, generation) performance, while on some other intervals negatively correlated to inference performance. Therefore, the design of the training schedule is rather putting more emphasis on the positively correlated regions, even if this would sacrifice the loss in some other regions. A direct evidence is shown in Table G.3.4 in Appx. G.3, which demonstrates that our schedule improves inference performance without memorizing/overfitting training data indeed.

## B    ALTERNATIVE FORMULATIONS OF DIFFUSION MODELS

Besides the *score prediction* target introduced in Sec. 2, diffusion models also adopt other prediction targets. The formulation of Eq. (2) motivates the *noise prediction* target (Ho et al., 2020). Let $\boldsymbol{\epsilon}_{\boldsymbol{\theta}}(\mathbf{x}_t, t) := -\sigma_t \mathbf{s}_{\boldsymbol{\theta}}(\mathbf{x}_t, t)$, the loss becomes

$$J(\boldsymbol{\theta}) = \mathbb{E}_{p(t)} w_t^{(\boldsymbol{\epsilon})} \mathbb{E}_{p_0(\mathbf{x}_0)} \mathbb{E}_{p(\boldsymbol{\epsilon})} \|\boldsymbol{\epsilon}_{\boldsymbol{\theta}}(\alpha_t \mathbf{x}_0 + \sigma_t \boldsymbol{\epsilon}, t) - \boldsymbol{\epsilon}\|^2, \quad \text{where } w_t^{(\boldsymbol{\epsilon})} = w_t^{(\mathbf{s})}/\sigma_t^2.$$

This formulation poses a friendly, bounded-scale learning target and avoids the artifact at $t = 0$ of the denoising score matching loss. If formally solving $\mathbf{x}_0$ from Eq. (1) and let

$$\mathbf{x}_{0\boldsymbol{\theta}}(\mathbf{x}_t, t) := \frac{\mathbf{x}_t - \sigma_t \boldsymbol{\epsilon}_{\boldsymbol{\theta}}(\mathbf{x}_t, t)}{\alpha_t} = \frac{\mathbf{x}_t + \sigma_t^2 \mathbf{s}_{\boldsymbol{\theta}}(\mathbf{x}_t, t)}{\alpha_t},$$

then we get the total loss becomes $J(\boldsymbol{\theta}) = \mathbb{E}_{p(t)} w_t^{(\mathbf{x}_0)} J_t^{(\mathbf{x}_0)}(\boldsymbol{\theta})$, where:
$J_t^{(\mathbf{x}_0)}(\boldsymbol{\theta}) := \mathbb{E}_{p(\mathbf{x}_0)} \mathbb{E}_{p(\epsilon)} \|\mathbf{x}_{0\boldsymbol{\theta}}(\alpha_t \mathbf{x}_0 + \sigma_t \boldsymbol{\epsilon}, t) - \mathbf{x}_0\|^2 = (\sigma_t^4/\alpha_t^2) J_t^{(\mathbf{s})}(\boldsymbol{\theta}), \quad w_t^{(\mathbf{x}_0)} = (\alpha_t^2/\sigma_t^4) w_t^{(\mathbf{s})}.$
It holds the semantics of *clean-data prediction* (Kingma et al., 2021; Karras et al., 2022), and can be viewed as denoising auto-encoders (Vincent et al., 2008; Alain and Bengio, 2014) with multiple noise scales. From the equivalent deterministic process, one can also derive the *vector-field prediction* target by

$$\mathbf{v}_{\boldsymbol{\theta}}(\mathbf{x}_t, t) := a_t \mathbf{x}_t - \frac{1}{2} g_t^2 \mathbf{s}_{\boldsymbol{\theta}}(\mathbf{x}_t, t),$$

an the loss function becomes $J(\boldsymbol{\theta}) = \mathbb{E}_{p(t)} w_t^{(\mathbf{v})} J_t^{(\mathbf{v})}(\boldsymbol{\theta})$, where

$$J_t^{(\mathbf{v})}(\boldsymbol{\theta}) := \mathbb{E}_{p(\mathbf{x}_0)}\mathbb{E}_{p(\epsilon)}\|\mathbf{v}_{\boldsymbol{\theta}}(\alpha_t\mathbf{x}_0 + \sigma_t\boldsymbol{\epsilon}, t) - (\alpha_t'\mathbf{x}_0 + \sigma_t'\boldsymbol{\epsilon})\|^2, \quad w_t^{(\mathbf{v})} = (4/g_t^4)w_t^{(\mathbf{s})}.$$

It coincides with velocity prediction (Salimans and Ho, 2022) and the flow matching formulation (Lipman et al., 2023; Liu et al., 2023): $\mathbf{v} := \alpha_t'\mathbf{x}_0 + \sigma_t'\boldsymbol{\epsilon}$ is the conditional vector field given $\mathbf{x}_0$ and $\boldsymbol{\epsilon}$ (same distribution as $\mathbf{x}_T$). In particular, if we set $\alpha_t = 1 - t$, $\sigma_t = t$, $T = 1$, then $a_t = \frac{1}{t-1}, \frac{1}{2}g_t^2 = \frac{t}{1-t}$, and we have $\mathbf{v} = \boldsymbol{\epsilon} - \mathbf{x}_0$, which corresponds to the Flow Matching (Lipman et al., 2023; Liu et al., 2023) formulation that is also widely used in generative modeling recently due to its simplicity.

## C  OPTIMAL SOLUTION OF DIFFUSION MODELS

In this section, we provide a detailed derivation of the optimal solution for diffusion models across various formulations. We demonstrate that in every case the models' learning targets are conditional expectations, as discussed in Sec. 3.1. To this end, we begin by introducing a useful lemma that outlines a key property of conditional expectations (Durrett, 2019).

**Lemma 3.** *Let $\mathbf{x}, \mathbf{y}$ be random vectors. Then the optimal approximation of $\mathbf{y}$ based on $\mathbf{x}$ is*

$$\mathbf{f}^*(\mathbf{x}) = \arg\min_{\mathbf{f}:\mathbb{R}^d \to \mathbb{R}^d} \mathbb{E}\|\mathbf{y} - \mathbf{f}(\mathbf{x})\|^2 = \mathbb{E}[\mathbf{y}|\mathbf{x}].$$

*Proof.* We can compute $\mathbb{E}\|\mathbf{y} - \mathbf{f}(\mathbf{x})\|^2$ directly by

$$\mathbb{E}\|\mathbf{y} - \mathbf{f}(\mathbf{x})\|^2 = \mathbb{E}\|\mathbf{y} - \mathbb{E}[\mathbf{y}|\mathbf{x}] + \mathbb{E}[\mathbf{y}|\mathbf{x}] - \mathbf{f}(\mathbf{x})\|^2$$
$$= \mathbb{E}\|\mathbf{y} - \mathbb{E}[\mathbf{y}|\mathbf{x}]\|^2 + \mathbb{E}\left[\|\mathbb{E}[\mathbf{y}|\mathbf{x}] - \mathbf{f}(\mathbf{x})\|^2\right] + 2\mathbb{E}\langle\mathbf{y} - \mathbb{E}[\mathbf{y}|\mathbf{x}], \mathbb{E}[\mathbf{y}|\mathbf{x}] - \mathbf{f}(\mathbf{x})\rangle.$$

Since

$$\mathbb{E}\langle\mathbf{y} - \mathbb{E}[\mathbf{y}|\mathbf{x}], \mathbb{E}[\mathbf{y}|\mathbf{x}] - \mathbf{f}(\mathbf{x})\rangle = \mathbb{E}\left[\mathbb{E}\langle\mathbf{y} - \mathbb{E}[\mathbf{y}|\mathbf{x}], \mathbb{E}[\mathbf{y}|\mathbf{x}] - \mathbf{f}(\mathbf{x})\rangle|\mathbf{x}\right] = 0,$$

we have the following decomposition:

$$\mathbb{E}\|\mathbf{y} - \mathbf{f}(\mathbf{x})\|^2 = \mathbb{E}\|\mathbf{y} - \mathbb{E}[\mathbf{y}|\mathbf{x}]\|^2 + \mathbb{E}\left[\|\mathbb{E}[\mathbf{y}|\mathbf{x}] - \mathbf{f}(\mathbf{x})\|^2\right].$$

Since $\mathbb{E}\left[\|\mathbb{E}[\mathbf{y}|\mathbf{x}] - \mathbf{f}(\mathbf{x})\|^2\right] \geqslant 0$, we have

$$\mathbb{E}\|\mathbf{y} - \mathbf{f}(\mathbf{x})\|^2 = \mathbb{E}\|\mathbf{y} - \mathbb{E}[\mathbf{y}|\mathbf{x}]\|^2 + \mathbb{E}\left[\|\mathbb{E}[\mathbf{y}|\mathbf{x}] - \mathbf{f}(\mathbf{x})\|^2\right]$$
$$\geqslant \mathbb{E}\|\mathbf{y} - \mathbb{E}[\mathbf{y}|\mathbf{x}]\|^2.$$

The inequality becomes equality if and only if $\mathbf{f}(\mathbf{x}) = \mathbb{E}[\mathbf{y}|\mathbf{x}]$. So the the optimal approximation of $\mathbf{y}$ based on $\mathbf{x}$ is $\mathbb{E}[\mathbf{y}|\mathbf{x}]$, *i.e.*,

$$\mathbf{f}^*(\mathbf{x}) = \arg\min_{\mathbf{f}:\mathbb{R}^d \to \mathbb{R}^d} \mathbb{E}\|\mathbf{y} - \mathbf{f}(\mathbf{x})\|^2 = \mathbb{E}[\mathbf{y}|\mathbf{x}].$$

$\square$

**Score prediction target.**  The score prediction target of diffusion model is given by

$$J_t^{(\mathbf{s})}(\boldsymbol{\theta}) := \mathbb{E}_{p(t)} w_t^{(\mathbf{s})} \mathbb{E}_{p_0(\mathbf{x}_0)} \mathbb{E}_{p(\mathbf{x}_t|\mathbf{x}_0)} \|\mathbf{s}_{\boldsymbol{\theta}}(\mathbf{x}_t, t) - \nabla_{\mathbf{x}_t} \log p(\mathbf{x}_t \mid \mathbf{x}_0)\|^2.$$

Then by Lemma 3, for any $t$ satisfying $w_t^{(\mathbf{s})} > 0$, the optimal solution of the network $\mathbf{s}_{\boldsymbol{\theta}}(\cdot, t) : \mathbb{R}^d \to \mathbb{R}^d$ is given by

$$\mathbf{s}_{\boldsymbol{\theta}}^*(\mathbf{x}_t, t) = \mathbb{E}_{p(\mathbf{x}_0|\mathbf{x}_t)}[\nabla_{\mathbf{x}_t} \log p(\mathbf{x}_t \mid \mathbf{x}_0)].$$

**Noise prediction target.**  The noise prediction target of diffusion model is given by

$$J_t^{(\boldsymbol{\epsilon})}(\boldsymbol{\theta}) := \mathbb{E}_{p(t)} w_t^{(\boldsymbol{\epsilon})} \mathbb{E}_{p_0(\mathbf{x}_0)} \mathbb{E}_{p(\boldsymbol{\epsilon})} \|\boldsymbol{\epsilon}_{\boldsymbol{\theta}}(\alpha_t\mathbf{x}_0 + \sigma_t\boldsymbol{\epsilon}, t) - \boldsymbol{\epsilon}\|^2.$$

Then by Lemma 3, for any $t$ satisfying $w_t^{(\boldsymbol{\epsilon})} > 0$, the optimal solution of the network $\boldsymbol{\epsilon}_{\boldsymbol{\theta}}(\cdot, t) : \mathbb{R}^d \to \mathbb{R}^d$ is given by

$$\boldsymbol{\epsilon}_{\boldsymbol{\theta}}^*(\mathbf{x}_t, t) = \mathbb{E}_{p(\boldsymbol{\epsilon}|\mathbf{x}_t)}[\boldsymbol{\epsilon}].$$

**Clean-data prediction target.**  The clean-data prediction target of diffusion model is given by

$$J_t^{(\mathbf{x}_0)}(\boldsymbol{\theta}) := \mathbb{E}_{p(\mathbf{x}_0)} \mathbb{E}_{p(\epsilon)} \|\mathbf{x}_{0\boldsymbol{\theta}}(\alpha_t\mathbf{x}_0 + \sigma_t\boldsymbol{\epsilon}, t) - \mathbf{x}_0\|^2.$$

Then by Lemma 3, for any $t$ satisfying $w_t^{(\mathbf{x}_0)} > 0$, the optimal solution of the network $\mathbf{x}_{0\boldsymbol{\theta}}(\cdot, t) : \mathbb{R}^d \to \mathbb{R}^d$ is given by

$$\mathbf{x}_{0\boldsymbol{\theta}}^*(\mathbf{x}_t, t) = \mathbb{E}_{p(\mathbf{x}_0|\mathbf{x}_t)}[\mathbf{x}_0].$$

**Vector-field prediction target.** The vector field prediction target of diffusion model is given by

$$J_t^{(\mathbf{v})}(\boldsymbol{\theta}) := \mathbb{E}_{p(\mathbf{x}_0)}\mathbb{E}_{p(\epsilon)}\|\mathbf{v}_{\boldsymbol{\theta}}(\alpha_t\mathbf{x}_0 + \sigma_t\boldsymbol{\epsilon}, t) - (\alpha_t'\mathbf{x}_0 + \sigma_t'\boldsymbol{\epsilon})\|^2.$$

Then by Lemma 3, for any $t$ satisfying $w_t^{(\mathbf{v})} > 0$, the optimal solution of the network $\mathbf{v}_{\boldsymbol{\theta}}(\cdot, t) : \mathbb{R}^d \to \mathbb{R}^d$ is given by

$$\mathbf{v}_{\boldsymbol{\theta}}^*(\mathbf{x}_t, t) = \mathbb{E}_{p(\mathbf{x}_0, \boldsymbol{\epsilon}|\mathbf{x}_t)}[\alpha_t'\mathbf{x}_0 + \sigma_t'\boldsymbol{\epsilon}].$$

## D  DETAILS ON THE GENERAL DIFFUSION FORMULATION

In this section, we introduce the detailed conversion between different diffusion formulations. As we have mentioned in Sec. 3.4, we can convert previous schedules to EDM formulation. The loss function in EDM formulation is given by

$$J(\theta) = \mathbb{E}_{p(\sigma)}w_\sigma\mathbb{E}_{p(\mathbf{x}_0), p(\epsilon)}\|\mathbf{x}_{0\boldsymbol{\theta}}(\mathbf{x}_0 + \sigma\boldsymbol{\epsilon}, \sigma) - \mathbf{x}_0\|^2,$$

where the denoiser has precondition $\mathbf{x}_{0\boldsymbol{\theta}}(\mathbf{x}, \sigma) = c_\sigma^{\text{skip}}\mathbf{x} + c_\sigma^{\text{out}}\mathbf{F}_\theta(c_\sigma^{\text{in}}\mathbf{x}, c_\sigma^{\text{noise}})$.

### D.1  CONVERT VP SCHEDULES TO EDM FORMULATION

**DDPM (VP), $\epsilon$-prediction loss function.** The loss function in DDPM formulation (Ho et al., 2020; Song et al., 2021b) is given by

$$J_{\text{DDPM}}^{(\epsilon)} = \mathbb{E}_{p^{\text{DDPM}}(t)}\mathbb{E}_{p(\mathbf{x}_0), p(\epsilon)}\|\boldsymbol{\epsilon}_\theta(\alpha_t\mathbf{x}_0 + \sigma_t\boldsymbol{\epsilon}, (M-1)t) - \boldsymbol{\epsilon}\|^2,$$

where $M = 1000$, $p^{\text{DDPM}}(t) = \mathcal{U}(\varepsilon_t, 1)$, $\varepsilon_t = 10^{-5}$, and

$$\sigma_t = \sqrt{1 - \exp(-(\beta_0 t + \frac{1}{2}(\beta_{\max} - \beta_0)t^2))},$$

$$\alpha_t = \sqrt{1 - \sigma^2} = \exp(-(\beta_0 t + \frac{1}{2}(\beta_{\max} - \beta_0)t^2)).$$

The diffusion process satisfying $\alpha_t^2 + \sigma_t^2 = 1$ is also known as the variance preserving (VP) process. In order to convert the DDPM formulation to the EDM formulation, we should transform the diffusion process to VE. Dividing $\sqrt{1 - \sigma_t^2}$ in both side of the equation $\mathbf{x}_t = \sqrt{1 - \sigma_t^2}\mathbf{x}_0 + \sigma_t\boldsymbol{\epsilon}$, we have

$$\frac{\mathbf{x}_t}{\sqrt{1 - \sigma_t^2}} = \mathbf{x}_0 + \frac{\sigma_t}{\sqrt{1 - \sigma_t^2}}\boldsymbol{\epsilon}.$$

Let $\hat{\mathbf{x}}_t = \frac{\mathbf{x}_t}{\sqrt{1-\sigma_t^2}}$ and $\hat{\sigma}_t = \frac{\sigma_t}{\sqrt{1-\sigma_t^2}}$, then $\hat{\mathbf{x}}_t = \mathbf{x}_0 + \hat{\sigma}_t\boldsymbol{\epsilon}$, *i.e.*, $\hat{\mathbf{x}}_t$, is a VE process. The inverse transform is given by $\sigma_t = \frac{\hat{\sigma}_t}{\sqrt{1+\hat{\sigma}_t^2}}$. Under this transformation, the loss function of DDPM becomes

$$J_{\text{DDPM}}^{(\epsilon)} = \mathbb{E}_{p^{\text{DDPM}}(t)}\mathbb{E}_{p(\mathbf{x}_0), p(\epsilon)}\left\|\boldsymbol{\epsilon}_\theta(\sqrt{1-\sigma_t^2}\mathbf{x}_0 + \sigma_t\boldsymbol{\epsilon}, (M-1)t) - \boldsymbol{\epsilon}\right\|^2$$

$$= \mathbb{E}_{p^{\text{DDPM}}(t)}\mathbb{E}_{p(\mathbf{x}_0), p(\epsilon)}\left\|\boldsymbol{\epsilon}_\theta(\mathbf{x}_t, (M-1)t) - \frac{\mathbf{x}_t - \sqrt{1-\sigma_t^2}\mathbf{x}_0}{\sigma_t}\right\|^2$$

$$= \mathbb{E}_{p^{\text{DDPM}}(t)}\mathbb{E}_{p(\mathbf{x}_0), p(\epsilon)}\left(\frac{1-\sigma_t^2}{\sigma_t^2}\right)\left\|\frac{\mathbf{x}_t}{\sqrt{1-\sigma_t^2}} - \frac{\sigma_t}{\sqrt{1-\sigma_t^2}}\boldsymbol{\epsilon}_\theta(\mathbf{x}_t, (M-1)t) - \mathbf{x}_0\right\|^2$$

$$= \mathbb{E}_{p^{\text{DDPM}}(t)}\mathbb{E}_{p(\mathbf{x}_0), p(\epsilon)}\frac{1}{\hat{\sigma}_t^2}\left\|\hat{\mathbf{x}}_t - \hat{\sigma}_t\boldsymbol{\epsilon}_\theta\left(\frac{\hat{\mathbf{x}}_t}{\sqrt{1+\hat{\sigma}_t^2}}, (M-1)t\right) - \mathbf{x}_0\right\|^2,$$

where we have used the relations $\hat{x}_t = \frac{x_t}{\sqrt{1-\sigma_t^2}}$ and $\sigma_t = \frac{\hat{\sigma}_t}{\sqrt{1+\hat{\sigma}_t^2}}$ to get the last equation. Comparing the loss function with the general loss function of EDM,

$$J_{\text{DDPM}}^{(\boldsymbol{\epsilon})} = \mathbb{E}_{p^{\text{DDPM}}(\hat{\sigma})} w^{\text{DDPM}}(\hat{\sigma}) \mathbb{E}_{p(\mathbf{x}_0), p(\boldsymbol{\epsilon})} \|\mathbf{x}_{0\boldsymbol{\theta}}(\mathbf{x}_0 + \hat{\sigma}\boldsymbol{\epsilon}, \hat{\sigma}) - \mathbf{x}_0\|^2,$$

$$\text{where} \quad \mathbf{x}_{0\boldsymbol{\theta}}(\mathbf{x}, \hat{\sigma}) = c_{\text{skip}}^{\text{DDPM}}(\hat{\sigma})\mathbf{x} + c_{\text{out}}^{\text{DDPM}}(\hat{\sigma})\boldsymbol{\epsilon}_\theta(c_{\text{in}}^{\text{DDPM}}(\hat{\sigma})\mathbf{x}, c_{\text{noise}}^{\text{DDPM}}(\hat{\sigma})),$$

we get

$$c_{\text{skip}}^{\text{DDPM}}(\hat{\sigma}_t) = 1, \qquad\qquad c_{\text{out}}^{\text{DDPM}}(\hat{\sigma}_t) = -\hat{\sigma}_t,$$

$$c_{\text{in}}^{\text{DDPM}}(\hat{\sigma}_t) = \frac{1}{\sqrt{1+\hat{\sigma}_t^2}}, \qquad\qquad c_{\text{noise}}^{\text{DDPM}}(\hat{\sigma}_t) = (M-1)t.$$

And the training schedule is given by

$$w_{\hat{\sigma}}^{\text{DDPM}} = \frac{1}{\hat{\sigma}^2}, \quad p^{\text{DDPM}}(\hat{\sigma}) = \left(\frac{\sigma}{\sqrt{1-\sigma^2}}\right)_{\#} \sigma_{t\#} \mathcal{U}(\varepsilon_t, 1).$$

### D.2 CONVERT VE SCHEDULES TO EDM FORMULATION

**Review of EDM, F-prediction loss function.** EDM Karras et al. (2022) proposes the "unit variance principle" to derive the EDM precondition. Recall that the denoiser has precondition $\mathbf{x}_{0\boldsymbol{\theta}}(\mathbf{x}, \sigma) = c_\sigma^{\text{skip}}\mathbf{x} + c_\sigma^{\text{out}}\mathbf{F}_\theta(c_\sigma^{\text{in}}\mathbf{x}, c^{\text{noise}})$, where $\mathbf{F}_\theta$ is the neural network, then the effective loss function is given by

$$J_{\text{EDM}}^{(\mathbf{F})}(\theta) = \mathbb{E}_{p^{\text{EDM}}(\sigma)} w^{\text{EDM}}(\sigma) \mathbb{E}_{p(\mathbf{x}_0), p(\boldsymbol{\epsilon})} \|\mathbf{x}_{0\boldsymbol{\theta}}(\mathbf{x}_0 + \sigma\boldsymbol{\epsilon}, \sigma) - \mathbf{x}_0\|^2$$

$$= \mathbb{E}_{p^{\text{EDM}}(\sigma)} w^{\text{EDM}}(\sigma) c_\sigma^{\text{out}\,2} \mathbb{E}_{p(\mathbf{x}_0), p(\boldsymbol{\epsilon})} \left\|\mathbf{F}_\theta(c_\sigma^{\text{in}}\mathbf{x}_\sigma, c^{\text{noise}}) - \frac{\mathbf{x}_0 - c_\sigma^{\text{skip}}\mathbf{x}_\sigma}{c_\sigma^{\text{out}}}\right\|^2,$$

where $\mathbf{x}_\sigma = \mathbf{x}_0 + \sigma\boldsymbol{\epsilon}$. The unit variance principle is given by

$$\text{Var}(c_\sigma^{\text{in}}\mathbf{x}_\sigma) = 1,$$

$$\text{Var}\left(\frac{\mathbf{x}_0 - c_\sigma^{\text{skip}}\mathbf{x}_\sigma}{c_\sigma^{\text{out}}}\right) = 1,$$

$$w^{\text{EDM}}(\sigma)c_\sigma^{\text{out}\,2} = 1.$$

Then we get the explicit expression of the precondition as follows:

$$c_{\text{skip}}^{\text{EDM}}(\sigma) = \frac{\sigma_{\text{data}}^2}{\sigma^2 + \sigma_{\text{data}}^2}, \qquad\qquad c_{\text{out}}^{\text{EDM}}(\sigma) = \frac{\sigma \cdot \sigma_{\text{data}}}{\sqrt{\sigma^2 + \sigma_{\text{data}}^2}},$$

$$c_{\text{in}}^{\text{EDM}}(\sigma) = \frac{1}{\sqrt{\sigma^2 + \sigma_{\text{data}}^2}}, \qquad\qquad c_{\text{noise}}^{\text{EDM}}(\sigma) = \frac{1}{4}\ln\sigma.$$

And the training schedule is given by

$$w_\sigma^{\text{EDM}} = \frac{\sigma^2 + \sigma_{\text{data}}^2}{(\sigma \cdot \sigma_{\text{data}})^2}, \quad p^{\text{EDM}}(\sigma) = \exp_{\#} \mathcal{N}(P_{\text{mean}}, P_{\text{std}}^2).$$

Lu and Song (2024) shows that when $\sigma_t = \sin(\frac{\pi}{2}t)$, the EDM loss function is equivalent to **v**-prediction of VP process.

**NCSN (VE), $\boldsymbol{\epsilon}$-prediction loss function.** The loss function in NCSN (VE) formulation (Song et al., 2021b) is given by

$$J_{\text{NCSN}}^{(\boldsymbol{\epsilon})} = \mathbb{E}_{p^{\text{NCSN}}(\sigma)} \mathbb{E}_{p(\mathbf{x}_0), p(\boldsymbol{\epsilon})} \left\|\boldsymbol{\epsilon}_\theta\left(\mathbf{x}_0 + \sigma\boldsymbol{\epsilon}, \ln\left(\frac{\sigma}{2}\right)\right) + \boldsymbol{\epsilon}\right\|^2,$$

where $p^{\text{NCSN}}(\sigma) = \exp_{\#} \mathcal{U}(\ln\sigma_{\min}, \ln\sigma_{\max})$, *i.e.*, $\ln\sigma \sim \mathcal{U}(\ln\sigma_{\min}, \ln\sigma_{\max})$. Then we can convert it to the EDM formulation:

$$J_{\text{NCSN}}^{(\boldsymbol{\epsilon})} = \mathbb{E}_{p^{\text{NCSN}}(\sigma)} \mathbb{E}_{p(\mathbf{x}_0), p(\boldsymbol{\epsilon})} \left\|\boldsymbol{\epsilon}_\theta\left(\mathbf{x}_0 + \sigma\boldsymbol{\epsilon}, \ln\left(\frac{\sigma}{2}\right)\right) + \boldsymbol{\epsilon}\right\|^2$$

$$= \mathbb{E}_{p^{\text{NCSN}}(\sigma)} \mathbb{E}_{p(\mathbf{x}_0), p(\boldsymbol{\epsilon})} \left\|\boldsymbol{\epsilon}_\theta\left(\mathbf{x}_0 + \sigma\boldsymbol{\epsilon}, \ln\left(\frac{\sigma}{2}\right)\right) + \frac{\mathbf{x}_\sigma - \mathbf{x}_0}{\sigma}\right\|^2$$

$$= \mathbb{E}_{p^{\mathrm{NCSN}}(\sigma)} \mathbb{E}_{p(\mathbf{x}_0), p(\boldsymbol{\epsilon})} \frac{1}{\sigma^2} \left\| \mathbf{x}_\sigma + \sigma \boldsymbol{\epsilon}_\theta \left( \mathbf{x}_0 + \sigma \boldsymbol{\epsilon}, \ln \left( \frac{\sigma}{2} \right) \right) - \mathbf{x}_0 \right\|^2.$$

Then we get the explicit expression of the precondition as follows:

$$c_{\mathrm{skip}}^{\mathrm{NCSN}}(\sigma) = 1, \qquad\qquad c_{\mathrm{out}}^{\mathrm{NCSN}}(\sigma) = \sigma,$$

$$c_{\mathrm{in}}^{\mathrm{NCSN}}(\sigma) = 1, \qquad\qquad c_{\mathrm{noise}}^{\mathrm{NCSN}}(\sigma) = \ln \left( \frac{\sigma}{2} \right).$$

And the training schedule is given by

$$w_\sigma^{\mathrm{NCSN}} = \frac{1}{\sigma^2}, \quad p^{\mathrm{NCSN}}(\sigma) = \exp_\# \mathcal{U}(\ln \sigma_{\min}, \ln \sigma_{\max}).$$

## D.3 Convert Flow Matching Schedules to EDM Formulation

**Flow Matching, v-prediction loss function.** In the original Flow Matching paper (Lipman et al., 2023), $p_t(\mathbf{x}_t)$ is the noise distribution when $t = 0$ and becomes the data distribution when $t = 1$. To align with the time line of diffusion models, we revert the original Flow Matching construction, *i.e.*, $p_t(\mathbf{x}_t)$ is the data distribution when $t = 0$ and becomes the noise distribution when $t = 1$. Then the loss function in the Flow Matching formulation is given by

$$J_{\mathrm{FM}}^{(\mathbf{v})}(\theta) = \mathbb{E}_{p^{\mathrm{FM}}(t)} \mathbb{E}_{p(\mathbf{x}_0), p(\boldsymbol{\epsilon})} \left\| \mathbf{v}_\theta \left( \alpha_t \mathbf{x}_0 + \sigma_t \boldsymbol{\epsilon}, t \right) - (\boldsymbol{\epsilon} - \mathbf{x}_0) \right\|^2,$$

where $\alpha_t = 1 - t, \sigma_t = t, p^{\mathrm{FM}}(t) = \mathcal{U}(0, 1)$. In order to convert the Flow Matching formulation to the EDM formulation, we should transform the flow matching diffusion process to the VE diffusion process. Dividing $(1 - \sigma_t)$ in both side of the equation $\mathbf{x}_t = (1 - \sigma_t)\mathbf{x}_0 + \sigma_t \boldsymbol{\epsilon}$, we have

$$\frac{\mathbf{x}_t}{1 - \sigma_t} = \mathbf{x}_0 + \frac{\sigma_t}{1 - \sigma_t} \boldsymbol{\epsilon}.$$

Let $\hat{\mathbf{x}}_t = \frac{\mathbf{x}_t}{1 - \sigma_t}, \hat{\sigma}_t = \frac{\sigma_t}{1 - \sigma_t}$, then $\hat{\mathbf{x}}_t$ satisfies $\hat{\mathbf{x}}_t = \mathbf{x}_0 + \hat{\sigma}_t \boldsymbol{\epsilon}$, *i.e.*, $\hat{\mathbf{x}}_t$ is a VE process. The inverse transform is given by $\sigma_t = \frac{\hat{\sigma}_t}{1 + \hat{\sigma}_t}$. Under this transformation, the loss function of Flow Matching becomes

$$J_{\mathrm{FM}}^{(\mathbf{v})}(\theta) = \mathbb{E}_{p^{\mathrm{FM}}(\sigma_t)} \mathbb{E}_{p(\mathbf{x}_0), p(\boldsymbol{\epsilon})} \| \mathbf{v}_\theta(\mathbf{x}_t, \sigma_t) - (\boldsymbol{\epsilon} - \mathbf{x}_0) \|^2$$

$$= \mathbb{E}_{p^{\mathrm{FM}}(\sigma_t)} \mathbb{E}_{p(\mathbf{x}_0), p(\boldsymbol{\epsilon})} \| \mathbf{v}_\theta(\mathbf{x}_t, \sigma_t) - \frac{\mathbf{x}_t - \mathbf{x}_0}{\sigma_t} \|^2$$

$$= \mathbb{E}_{p^{\mathrm{FM}}(\sigma_t)} \mathbb{E}_{p(\mathbf{x}_0), p(\boldsymbol{\epsilon})} \frac{1}{\sigma_t^2} \| \mathbf{x}_t - \sigma_t \mathbf{v}_\theta(\mathbf{x}_t, \sigma_t) - \mathbf{x}_0 \|^2$$

$$= \mathbb{E}_{p^{\mathrm{FM}}(\hat{\sigma})} \mathbb{E}_{p(\mathbf{x}_0), p(\boldsymbol{\epsilon})} \left( \frac{1 + \hat{\sigma}_t}{\hat{\sigma}_t} \right)^2 \left\| \frac{\hat{\mathbf{x}}_t}{1 + \hat{\sigma}_t} - \frac{\hat{\sigma}_t}{1 + \hat{\sigma}_t} \mathbf{v}_\theta \left( \frac{\hat{\mathbf{x}}_t}{1 + \hat{\sigma}_t}, \frac{\hat{\sigma}_t}{1 + \hat{\sigma}_t} \right) - \mathbf{x}_0 \right\|^2,$$

where we use the relations $\hat{\mathbf{x}}_t = \frac{\mathbf{x}_t}{1 - \sigma_t}, \sigma_t = \frac{\hat{\sigma}_t}{1 + \hat{\sigma}_t}$ to get the last equation. Compare the loss function with the general loss function of EDM

$$J(\theta) = \mathbb{E}_{p(\sigma)} w_\sigma \mathbb{E}_{p(\mathbf{x}_0), p(\boldsymbol{\epsilon})} \| \mathbf{x}_{0\theta}(\mathbf{x}_0 + \sigma \boldsymbol{\epsilon}, \sigma) - \mathbf{x}_0 \|^2,$$

we get

$$c_{\mathrm{skip}}^{\mathrm{FM}}(\hat{\sigma}) = \frac{1}{1 + \hat{\sigma}}, \qquad\qquad c_{\mathrm{out}}^{\mathrm{FM}}(\hat{\sigma}) = -\frac{\hat{\sigma}}{1 + \hat{\sigma}},$$

$$c_{\mathrm{in}}^{\mathrm{FM}}(\hat{\sigma}) = \frac{1}{1 + \hat{\sigma}}, \qquad\qquad c_{\mathrm{noise}}^{\mathrm{FM}}(\hat{\sigma}) = \frac{\hat{\sigma}}{1 + \hat{\sigma}}.$$

And the training schedule is given by

$$w_{\hat{\sigma}}^{\mathrm{FM}} = \frac{(1 + \hat{\sigma})^2}{\hat{\sigma}^2}, \quad p^{\mathrm{FM}}(\hat{\sigma}) = \left( \frac{\sigma}{1 - \sigma} \right)_\# \mathcal{U}(0, 1).$$

**Stable Diffusion 3, v-prediction loss function.** The Stable Diffusion 3 framework (Esser et al., 2024) also uses the Flow Matching (Rectified Flow) diffusion process and constructs v-prediction loss function. The difference is that SD3 proposes the logit-normal noise schedule, *i.e.*,

$$p_{\ln}(t; m, s) = \frac{1}{s\sqrt{2\pi}} \frac{1}{t(1 - t)} \exp \left( -\frac{(\log \frac{t}{1-t} - m)^2}{2s^2} \right).$$

Esser et al. (2024) shows that $m = 0, s = 1$ consistently achieves good performance. Let $p^{\text{SD3}}(t) = p_{\ln}(t; 0, 1)$. Then the SD3 loss function is given by

$$J_{\text{SD3}}^{(\mathbf{v})}(\theta) = \mathbb{E}_{p^{\text{SD3}}(t)} \mathbb{E}_{p(\mathbf{x}_0), p(\boldsymbol{\epsilon})} \left\| \mathbf{v}_\theta \left( \alpha_t \mathbf{x}_0 + \sigma_t \boldsymbol{\epsilon}, t \right) - (\boldsymbol{\epsilon} - \mathbf{x}_0) \right\|^2.$$

It is obvious that the SD3 loss function has the same precondition with FM, *i.e.*,

$$c_{\text{skip}}^{\text{SD3}}(\hat{\sigma}) = \frac{1}{1 + \hat{\sigma}}, \qquad\qquad c_{\text{out}}^{\text{SD3}}(\hat{\sigma}) = -\frac{\hat{\sigma}}{1 + \hat{\sigma}},$$

$$c_{\text{in}}^{\text{SD3}}(\hat{\sigma}) = \frac{1}{1 + \hat{\sigma}}, \qquad\qquad c_{\text{noise}}^{\text{SD3}}(\hat{\sigma}) = \frac{\hat{\sigma}}{1 + \hat{\sigma}}.$$

Since $\hat{\sigma}_t = \frac{\sigma_t}{1 - \sigma_t} = \frac{t}{1 - t} = \text{logit}(t)$, then $p^{\text{SD3}}(\hat{\sigma}_t) = \mathcal{N}(0, 1)$. So the training schedule is given by

$$w_{\hat{\sigma}}^{\text{SD3}} = \frac{(1 + \hat{\sigma})^2}{\hat{\sigma}^2}, \quad p^{\text{SD3}}(\hat{\sigma}) = \exp_{\#} \mathcal{N}(0, 1).$$

## E PROOFS

### E.1 PROOF OF THEOREM 1

**Theorem 4.** *The optimal loss value for clean-data prediction defined in Eq.* (4) *is:*

$$J_t^{(\mathbf{x}_0)^*} = \underbrace{\mathbb{E}_{p(\mathbf{x}_0)} \|\mathbf{x}_0\|^2}_{=:A} - \underbrace{\mathbb{E}_{p(\mathbf{x}_t)} \left\| \mathbb{E}_{p(\mathbf{x}_0 | \mathbf{x}_t)} [\mathbf{x}_0] \right\|^2}_{=:B_t}, \quad J^* = \mathbb{E}_{p(t)} w_t^{(\mathbf{x}_0)} J_t^{(\mathbf{x}_0)^*}.$$

*Proof.* According to Appx. C, the optimal solution of the network for clean-data prediction is given by $\mathbf{x}_{0\theta}(\mathbf{x}_t, t) = \mathbb{E}[\mathbf{x}_0 \mid \mathbf{x}_t]$, where $\mathbf{x}_t = \alpha_t \mathbf{x}_0 + \sigma_t \boldsymbol{\epsilon}$. And the loss function is minimized if and only if $\mathbf{x}_{0\theta}(\mathbf{x}_t, t) = \mathbb{E}[\mathbf{x}_0 \mid \mathbf{x}_t]$, and the optimal loss value is given by $\mathbb{E}_{p(\mathbf{x}_0), p(\epsilon)} \|\mathbb{E}[\mathbf{x}_0 \mid \mathbf{x}_t] - \mathbf{x}_0\|^2$. So we have

$$J_t^{(\mathbf{x}_0)^*} = \mathbb{E}_{p(\mathbf{x}_0), p(\epsilon)} \|\mathbb{E}[\mathbf{x}_0 \mid \mathbf{x}_t] - \mathbf{x}_0\|^2$$

$$= \mathbb{E}_{p(\mathbf{x}_0)} \|\mathbf{x}_0\|^2 - \mathbb{E}_{p(\mathbf{x}_t)} \left\| \mathbb{E}_{p(\mathbf{x}_0 | \mathbf{x}_t)} [\mathbf{x}_0] \right\|^2,$$

and $J^* = \mathbb{E}_{p(t)} w_t^{(\mathbf{x}_0)} J_t^{(\mathbf{x}_0)^*}$. $\qquad\square$

### E.2 PROOF OF THEOREM 2

The following theorem is a detailed version of Theorem 2.

**Theorem 5.** *For a given a subset $\{\mathbf{x}_0^{(l)}\}_{l=1}^L$, the $\hat{B}_t^{\text{cDOL}}$ estimator given in Eq.* (11) *converges a.s. to the following expression as $M \to \infty$:*

$$\hat{B}_t^{\text{cDOL}} \to \mathbb{E}_{p(\boldsymbol{\epsilon})} \left[ \frac{1}{L} \sum_{i=1}^L \left\| \frac{\sum_{l \neq i}^L \mathbf{x}_0^{(l)} K_t(\alpha_t \mathbf{x}_0^{(i)} + \sigma_t \boldsymbol{\epsilon}, \mathbf{x}_0^{(l)}) + \frac{1}{C} \mathbf{x}_0^{(i)} K_t(\alpha_t \mathbf{x}_0^{(i)} + \sigma_t \boldsymbol{\epsilon}, \mathbf{x}_0^{(i)})}{\sum_{l \neq i}^L K_t(\alpha_t \mathbf{x}_0^{(i)} + \sigma_t \boldsymbol{\epsilon}, \mathbf{x}_0^{(l)}) + \frac{1}{C} K_t(\alpha_t \mathbf{x}_0^{(i)} + \sigma_t \boldsymbol{\epsilon}, \mathbf{x}_0^{(i)})} \right\|^2 \right].$$

*So $\hat{B}_t^{\text{cDOL}}$ is a consistent estimator, $\forall C > 0$. Furthermore, the $\hat{B}_t^{\text{cDOL}}$ estimator with subset size $L$ has the same expectation as the SNIS estimator $\hat{B}_t^{\text{SNIS}}$ with subset size $L - 1$ when $M \to \infty, C \to \infty$.*

*Proof.* The cDOL estimator is given by

$$\hat{B}_t^{\text{cDOL}} = \frac{1}{M} \sum_{\tilde{m}=1}^M X_{\tilde{m}}, \quad \text{where } X_{\tilde{m}} := \left\| \frac{\sum_{\substack{l=1, \\ l \neq l_{\tilde{m}}}}^L \mathbf{x}_0^{(l)} K_t(\mathbf{x}_t^{(\tilde{m})}, \mathbf{x}_0^{(l)}) + \frac{1}{C} \mathbf{x}_0^{(l_{\tilde{m}})} K_t(\mathbf{x}_t^{(\tilde{m})}, \mathbf{x}_0^{(l_{\tilde{m}})})}{\sum_{\substack{l'=1, \\ l' \neq l_{\tilde{m}}}}^L K_t(\mathbf{x}_t^{(\tilde{m})}, \mathbf{x}_0^{(l')}) + \frac{1}{C} K_t(\mathbf{x}_t^{(\tilde{m})}, \mathbf{x}_0^{(l_{\tilde{m}})})} \right\|^2.$$

Given a subset $\{\mathbf{x}_0^{(l)}\}_{l=1}^L$, since $\{l_{\tilde{m}}\}_{\tilde{m}=1}^M$ and $\{\boldsymbol{\epsilon}_{\tilde{m}}\}_{\tilde{m}=1}^M$ are i.i.d. respectively, so $\{\mathbf{x}_{\tilde{m}} = \alpha_t \mathbf{x}_0^{l_{\tilde{m}}} + \sigma_t \boldsymbol{\epsilon}_{\tilde{m}}\}_{\tilde{m}=1}^M$ are i.i.d. random vectors. Then $\{X_{\tilde{m}}\}_{\tilde{m}=1}^M$ are i.i.d. random variables. By the strong

law of large number (Durrett, 2019), the estimator $\hat{B}_t^{\text{cDOL}}$ converges almost surely to the following expression as $M \to \infty$: $\hat{B}_t^{\text{cDOL}} \overset{M\to\infty,\text{a.s.}}{\to} \mathbb{E} X$

$$
= \mathbb{E}_{\tilde{l},\boldsymbol{\epsilon}} \left\| \frac{\sum_{\substack{l=1,\\l\neq\tilde{l}}}^{L} \mathbf{x}_0^{(l)} K_t(\alpha_t\mathbf{x}_0^{(\tilde{l})} + \sigma_t\boldsymbol{\epsilon}, \mathbf{x}_0^{(l)}) + \frac{1}{C}\mathbf{x}_0^{(\tilde{l})} K_t(\alpha_t\mathbf{x}_0^{(\tilde{l})} + \sigma_t\boldsymbol{\epsilon}, \mathbf{x}_0^{(\tilde{l})})}{\sum_{\substack{l'=1,\\l'\neq\tilde{l}}}^{L} K_t(\alpha_t\mathbf{x}_0^{(\tilde{l})} + \sigma_t\boldsymbol{\epsilon}, \mathbf{x}_0^{(l')}) + \frac{1}{C}K_t(\alpha_t\mathbf{x}_0^{(\tilde{l})} + \sigma_t\boldsymbol{\epsilon}, \mathbf{x}_0^{(\tilde{l})})} \right\|^2
$$

$$
= \mathbb{E}_{\boldsymbol{\epsilon}} \left[ \frac{1}{L} \sum_{i=1}^{L} \left\| \frac{\sum_{\substack{l=1,\\l\neq i}}^{L} \mathbf{x}_0^{(l)} K_t(\alpha_t\mathbf{x}_0^{(i)} + \sigma_t\boldsymbol{\epsilon}, \mathbf{x}_0^{(l)}) + \frac{1}{C}\mathbf{x}_0^{(i)} K_t(\alpha_t\mathbf{x}_0^{(i)} + \sigma_t\boldsymbol{\epsilon}, \mathbf{x}_0^{(i)})}{\sum_{\substack{l'=1,\\l'\neq i}}^{L} K_t(\alpha_t\mathbf{x}_0^{(i)} + \sigma_t\boldsymbol{\epsilon}, \mathbf{x}_0^{(l')}) + \frac{1}{C}K_t(\alpha_t\mathbf{x}_0^{(i)} + \sigma_t\boldsymbol{\epsilon}, \mathbf{x}_0^{(i)})} \right\|^2 \right].
$$

This completes the first statement. Since

$$
\frac{1}{L-1} \sum_{\substack{l=1,\\l\neq l_{\tilde{m}}}}^{L} \mathbf{x}_0^{(l)} K_t(\mathbf{x}_t^{(\tilde{m})}, \mathbf{x}_0^{(l)}) \overset{L\to\infty,\text{a.s}}{\to} \mathbb{E}_{p(\mathbf{x}_0)}[\mathbf{x}_0 K_t(\mathbf{x}_t^{(\tilde{m})}, \mathbf{x}_0)],
$$

$$
\frac{1}{L-1} \sum_{\substack{l=1,\\l\neq l_{\tilde{m}}}}^{L} K_t(\mathbf{x}_t^{(\tilde{m})}, \mathbf{x}_0^{(l)}) \overset{L\to\infty,\text{a.s}}{\to} \mathbb{E}_{p(\mathbf{x}_0)}[K_t(\mathbf{x}_t^{(\tilde{m})}, \mathbf{x}_0)], \forall \tilde{m},
$$

$$
\frac{1}{C(L-1)} \mathbf{x}_0^{(i)} K_t(\alpha_t\mathbf{x}_0^{(i)} + \sigma_t\boldsymbol{\epsilon}, \mathbf{x}_0^{(i)}) \overset{L\to\infty,\text{a.s}}{\to} 0,
$$

$$
\frac{1}{C(L-1)} K_t(\alpha_t\mathbf{x}_0^{(i)} + \sigma_t\boldsymbol{\epsilon}, \mathbf{x}_0^{(i)}) \overset{L\to\infty,\text{a.s}}{\to} 0,
$$

then $X_{\tilde{m}} \overset{L\to\infty,\text{a.s}}{\to} \mathbb{E}_{p(\mathbf{x}_0|\mathbf{x}_t^{(\tilde{m})})}[\mathbf{x}_0] = \frac{\mathbb{E}_{p(\mathbf{x}_0)}[\mathbf{x}_0 K_t(\mathbf{x}_t^{(\tilde{m})}, \mathbf{x}_0)]}{\mathbb{E}_{p(\mathbf{x}_0)}[K_t(\mathbf{x}_t^{(\tilde{m})}, \mathbf{x}_0)]}$. So we have

$$
\hat{B}_t^{\text{cDOL}} \overset{M,L\to\infty,\text{a.s}}{\to} \mathbb{E}_{p(\mathbf{x}_t)} \left\| \mathbb{E}_{p(\mathbf{x}_0|\mathbf{x}_t)}[\mathbf{x}_0] \right\|^2, \quad \forall C > 0.
$$

Hence, $\hat{B}_t^{\text{cDOL}} \overset{M,L\to\infty,\mathbb{P}}{\to} \mathbb{E}_{p(\mathbf{x}_t)} \left\| \mathbb{E}_{p(\mathbf{x}_0|\mathbf{x}_t)}[\mathbf{x}_0] \right\|^2, \forall C > 0$, *i.e.*, $\hat{B}_t^{\text{cDOL}}$ is consistent.

The SNIS estimator with subset size $L-1$ is given by

$$
\hat{B}_t^{\text{SNIS}} := \frac{1}{M} \sum_{m=1}^{M} Y_m, \quad \text{where } Y_m := \left\| \frac{\sum_{l=1}^{L-1} \mathbf{x}_0^{(l)} K_t(\mathbf{x}_t^{(m)}, \mathbf{x}_0^{(l)})}{\sum_{l'=1}^{L-1} K_t(\mathbf{x}_t^{(m)}, \mathbf{x}_0^{(l')})} \right\|^2.
$$

Notice that $\{\mathbf{x}_t^{(m)}\}_{m=1}^{M}$ are i.i.d. random vectors, so $\{Y_m\}_{m=1}^{M}$ are i.i.d. random variables. So by the strong law of large number, $\hat{B}_t^{\text{SNIS}}$ converges to the following expressions almost surely:

$$
\hat{B}_t^{\text{SNIS}} \overset{M\to\infty,\text{a.s}}{\to} \mathbb{E} Y = \mathbb{E}_{\mathbf{x}_t} \left\| \frac{\sum_{l=1}^{L-1} \mathbf{x}_0^{(l)} K_t(\mathbf{x}_t, \mathbf{x}_0^{(l)})}{\sum_{l'=1}^{L-1} K_t(\mathbf{x}_t, \mathbf{x}_0^{(l')})} \right\|^2.
$$

Next, we take the expectation of the subset for the cDOL and the SNIS estimator, respectively. Since $\{\mathbf{x}_0^{(l)}\}_{l=1}^{L}$ are i.i.d. random vectors, the expectation of the cDOL can be simplified as:

$$\mathbb{E}_{\{\mathbf{x}_0^{(l)}\}_{l=1}^L} \hat{B}_t^{\text{cDOL}}$$

$$= \mathbb{E}_{\boldsymbol{\epsilon},\{\mathbf{x}_0^{(l)}\}_{l=1}^L} \frac{1}{L} \sum_{i=1}^L \left\| \frac{\sum\limits_{\substack{l=1,\\l\neq i}}^L \mathbf{x}_0^{(l)} K_t(\alpha_t \mathbf{x}_0^{(i)} + \sigma_t \boldsymbol{\epsilon}, \mathbf{x}_0^{(l)}) + \frac{1}{C}\mathbf{x}_0^{(i)} K_t(\alpha_t \mathbf{x}_0^{(i)} + \sigma_t \boldsymbol{\epsilon}, \mathbf{x}_0^{(i)})}{\sum\limits_{\substack{l'=1,\\l'\neq i}}^L K_t(\alpha_t \mathbf{x}_0^{(i)} + \sigma_t \boldsymbol{\epsilon}, \mathbf{x}_0^{(l')}) + \frac{1}{C} K_t(\alpha_t \mathbf{x}_0^{(i)} + \sigma_t \boldsymbol{\epsilon}, \mathbf{x}_0^{(i)})} \right\|^2$$

$$= \mathbb{E}_{\boldsymbol{\epsilon},\{\mathbf{x}_0^{(l)}\}_{l=1}^{L-1},\mathbf{x}_0^{(L)}} \left\| \frac{\sum\limits_{l=1}^{L-1} \mathbf{x}_0^{(l)} K_t(\alpha_t \mathbf{x}_0^{(L)} + \sigma_t \boldsymbol{\epsilon}, \mathbf{x}_0^{(l)}) + \frac{1}{C}\mathbf{x}_0^{(L)} K_t(\alpha_t \mathbf{x}_0^{(L)} + \sigma_t \boldsymbol{\epsilon}, \mathbf{x}_0^{(L)})}{\sum\limits_{l'=1}^{L-1} K_t(\alpha_t \mathbf{x}_0^{(L)} + \sigma_t \boldsymbol{\epsilon}, \mathbf{x}_0^{(l')}) + \frac{1}{C} K_t(\alpha_t \mathbf{x}_0^{(L)} + \sigma_t \boldsymbol{\epsilon}, \mathbf{x}_0^{(L)})} \right\|^2.$$

Notice that $\mathbf{x}_t = \alpha_t \mathbf{x}_0 + \sigma_t \boldsymbol{\epsilon}$, so when $C \to \infty$:

$$\mathbb{E}_{\{\mathbf{x}_0^{(l)}\}_{l=1}^L} \hat{B}_t^{\text{cDOL}} \overset{C\to\infty}{\Rightarrow} \mathbb{E}_{\boldsymbol{\epsilon},\mathbf{x}_0^{(L)},\{\mathbf{x}_0^l\}_{l=1}^{L-1}} \left\| \frac{\sum\limits_{l=1}^{L-1} \mathbf{x}_0^{(l)} K_t(\alpha_t \mathbf{x}_0^{(L)} + \sigma_t \boldsymbol{\epsilon}, \mathbf{x}_0^{(l)})}{\sum\limits_{l'=1}^{L-1} K_t(\alpha_t \mathbf{x}_0^{(L)} + \sigma_t \boldsymbol{\epsilon}, \mathbf{x}_0^{(l')})} \right\|^2$$

$$= \mathbb{E}_{\mathbf{x}_t,\{\mathbf{x}_0^{(l)}\}_{l=1}^{L-1}} \left\| \frac{\sum_{l=1}^{L-1} \mathbf{x}_0^{(l)} K_t(\mathbf{x}_t, \mathbf{x}_0^{(l)})}{\sum_{l'=1}^{L-1} K_t(\mathbf{x}_t, \mathbf{x}_0^{(l')})} \right\|^2.$$

The expectation of the SNIS estimator is given by

$$\mathbb{E}_{\{\mathbf{x}_0^{(l)}\}_{l=1}^L} \hat{B}_t^{\text{SNIS}} = \mathbb{E}_{\{\mathbf{x}_0^l\}_{l=1}^{L-1}} \mathbb{E}_{\mathbf{x}_t} Y = \mathbb{E}_{\mathbf{x}_t,\{\mathbf{x}_0^{(l)}\}_{l=1}^{L-1}} \left\| \frac{\sum_{l\in[L-1]} \mathbf{x}_0^{(l)} K_t(\mathbf{x}_t, \mathbf{x}_0^{(l)})}{\sum_{l'\in[L-1]} K_t(\mathbf{x}_t, \mathbf{x}_0^{(l')})} \right\|^2.$$

So we can conclude that the $\hat{B}_t^{\text{cDOL}}$ estimator with subset size $L$ has the same expectation as the SNIS estimator $\hat{B}_t^{\text{SNIS}}$ with subset size $L-1$ when $M \to \infty, C \to \infty$.

$\square$

## F  BACKGROUND ON IMPORTANCE SAMPLING

Assume $\mathbf{x}$ is a random variable, $\mathbf{f} : \mathbb{R}^d \to \mathbb{R}^d$ is a vector value function. Our goal is to estimate the expectation of $\mathbf{f}(\mathbf{x})$ under a given probability density function $\pi(\mathbf{x})$, that is,

$$\mathbf{I} = \int \mathbf{f}(\mathbf{x})\pi(\mathbf{x})\mathrm{d}\mathbf{x}.$$

However, if $\mathbf{f}(x)$ is significant primarily in regions where $\pi(x)$ is low, the standard Monte Carlo estimator may provide poor accuracy due to infrequent sampling of these regions—an issue often referred to as the rare event problem. Importance sampling is designed to address this challenge.

Note that

$$\mathbf{I} = \int \mathbf{f}(\mathbf{x}) \frac{\pi(\mathbf{x})}{q(\mathbf{x})} q(\mathbf{x})\mathrm{d}\mathbf{x} = \int \mathbf{f}(\mathbf{x}) w(\mathbf{x}) q(\mathbf{x})\mathrm{d}\mathbf{x},$$

where $w(\mathbf{x}) = \frac{\pi(\mathbf{x})}{q(\mathbf{x})}$ is the weight function and $q(\mathbf{x})$ is the probability density function of the proposal distribution. The importance sampling estimator is therefore given by

$$\hat{\mathbf{I}}_{\text{IS}} = \frac{1}{N} \sum_{i=1}^N \mathbf{f}(\mathbf{x}_i) w(\mathbf{x}_i), \quad \mathbf{x}_i \overset{\text{i.i.d.}}{\sim} q(\mathbf{x}).$$

The IS estimator is a powerful tool when both $\pi(x)$ and $q(x)$ are known exactly. However, when the densities are only known up to a normalizing constant (*i.e.*,, we can access only $\hat{\pi}(\mathbf{x}) = \frac{\pi(\mathbf{x})}{Z_\pi}$ and $\hat{q}(\mathbf{x}) = \frac{q(\mathbf{x})}{Z_q}$), the standard importance sampling estimator cannot be applied directly. In this case,

self-normalized importance sampling (SNIS) is used. Define $\hat{w}(\mathbf{x}) := \frac{\hat{\pi}(\mathbf{x})}{\hat{q}(\mathbf{x})}$; then

$$\mathbf{I} = \frac{\int \mathbf{f}(\mathbf{x})\hat{w}(\mathbf{x})q(\mathbf{x})\mathrm{d}\mathbf{x}}{\int \hat{w}(\mathbf{x})q(\mathbf{x})\mathrm{d}\mathbf{x}}.$$

The SNIS estimator is thus constructed as

$$\hat{\mathbf{I}}_{\text{SNIS}} = \frac{\sum_{i=1}^{N} \mathbf{f}(\mathbf{x}_i)\hat{w}(\mathbf{x}_i)}{\sum_{i=1}^{N} \hat{w}(\mathbf{x}_i)}, \quad \mathbf{x}_i \overset{\text{i.i.d.}}{\sim} q(\mathbf{x}).$$

We can see that the IS estimator is unbiased:

$$\begin{aligned}
\mathbb{E}\hat{\mathbf{I}}_{\text{IS}} &= \frac{1}{N} \sum_{i=1}^{N} \mathbb{E}[\mathbf{f}(\mathbf{x}_i)w(\mathbf{x}_i)] \\
&= \frac{1}{N} \sum_{i=1}^{N} \int \mathbf{f}(\mathbf{x}_i)w(\mathbf{x}_i)q(\mathbf{x}_i)\,\mathrm{d}\mathbf{x}_i \\
&= \int \mathbf{f}(\mathbf{x})\pi(\mathbf{x})\,\mathrm{d}\mathbf{x} = \mathbf{I}.
\end{aligned}$$

Unfortunately, the SNIS estimator is biased, but it is asymptotically unbiased and remains a consistent estimator Sanz-Alonso and Al-Ghattas (2024). The precise definitions are as follows.

**Definition 6.** Assume $\hat{\mathbf{I}}, \{\hat{\mathbf{I}}_n\}$ are estimators for $\mathbf{I}$. Then we have the following definitions:

- $\hat{\mathbf{I}}$ is said to be *unbiased* if $\mathbb{E}\hat{\mathbf{I}} = \mathbf{I}$.

- $\{\hat{\mathbf{I}}_n\}$ is said to be *asymptotically unbiased* if $\mathbb{E}\hat{\mathbf{I}}_n \to \mathbf{I}$ as $n \to \infty$.

**Definition 7.** Assume $\hat{\mathbf{I}}_n$ is the estimator for $\mathbf{I}$, $\forall n > 0$. Then $\hat{\mathbf{I}}_n$ is called *consistent* for $\mathbf{I}$ if $\hat{\mathbf{I}}_n \overset{\mathbb{P}}{\to} \mathbf{I}$ as $n \to \infty$.

**Proposition 8.** *Assume* $\|\mathbf{f}(\mathbf{x})\| \leqslant 1$, $\mathbb{E}_{q(\mathbf{x})}\hat{w}(\mathbf{x}) < \infty$, $\mathbb{E}_{q(\mathbf{x})}\hat{w}^2(\mathbf{x}) < \infty$, *then the following holds:*

1. $\mathbb{E}\|\hat{\mathbf{I}}_{SNIS} - \mathbf{I}\|^2 \leqslant \frac{4}{N} \frac{\mathbb{E}_{q(\mathbf{x})}[\hat{w}(\mathbf{x})^2]}{(\mathbb{E}_{q(\mathbf{x})}[\hat{w}(\mathbf{x})])^2}$;

2. $\|\mathbb{E}[(\hat{\mathbf{I}}_{SNIS} - \mathbf{I})]\| \leqslant \frac{2}{N} \frac{\mathbb{E}_{q(\mathbf{x})}[\hat{w}(\mathbf{x})^2]}{(\mathbb{E}_{q(\mathbf{x})}[\hat{w}(\mathbf{x})])^2}$.

*So we can conclude that the SNIS estimator is asymptotically unbiased.*

For completeness, we give the proof of the proposition. The proof is modified from Sanz-Alonso and Al-Ghattas (2024).

*Proof.* To simplify our notation, let

$$\hat{\mathbf{J}}_N = \sum_{i=1}^{N} \mathbf{f}(\mathbf{x}_i)\hat{w}(\mathbf{x}_i) \quad \hat{P}_N = \sum_{i=1}^{N} \hat{w}(\mathbf{x}_i), \quad \mathbf{x}_i \overset{\text{i.i.d.}}{\sim} q(\mathbf{x}).$$

Then $\hat{\mathbf{I}}_{\text{SNIS}} = \hat{\mathbf{J}}_N/\hat{P}_N$. Notice that

$$\begin{aligned}
\hat{\mathbf{I}}_{\text{SNIS}} - \mathbf{I} &= \hat{\mathbf{I}}_{\text{SNIS}} - \frac{\mathbb{E}_{q(\mathbf{x})}[\mathbf{f}(\mathbf{x})\hat{w}(\mathbf{x})]}{\mathbb{E}_{q(\mathbf{x})}[\hat{w}(\mathbf{x})]} \\
&= \frac{\hat{\mathbf{I}}_{\text{SNIS}}\mathbb{E}_{q(\mathbf{x})}[\hat{w}(\mathbf{x})] - \mathbb{E}_{q(\mathbf{x})}[\mathbf{f}(\mathbf{x})\hat{w}(\mathbf{x})]}{\mathbb{E}_{q(\mathbf{x})}[\hat{w}(\mathbf{x})]} \\
&= \frac{\hat{\mathbf{I}}_{\text{SNIS}}\left(\mathbb{E}_{q(\mathbf{x})}[\hat{w}(\mathbf{x})] - \hat{P}_N\right) - \left(\mathbb{E}_{q(\mathbf{x})}[\mathbf{f}(\mathbf{x})\hat{w}(\mathbf{x})] - \hat{\mathbf{J}}_N\right)}{\mathbb{E}_{q(\mathbf{x})}[\hat{w}(\mathbf{x})]}.
\end{aligned}$$

Since

$$\|\mathbf{x} - \mathbf{y}\|^2 \leqslant (\|\mathbf{x}\| + \|\mathbf{y}\|)^2 \leqslant 2(\|\mathbf{x}\|^2 + \|\mathbf{y}\|^2),$$

then we can use the identity to bound the variance of $\hat{\mathbf{I}}_{\text{SNIS}}$:

$$\mathbb{E}\|\hat{\mathbf{I}}_{\text{SNIS}} - \mathbf{I}\|^2$$

$$\leqslant \frac{2}{(\mathbb{E}_{q(\mathbf{x})}[\hat{w}(\mathbf{x})])^2} \left( \mathbb{E}\left[ \|\hat{\mathbf{I}}_{\text{SNIS}}\|^2 (\mathbb{E}_{q(\mathbf{x})}[\hat{w}(\mathbf{x})] - \hat{P}_N)^2 \right] + \mathbb{E}\left[ (\mathbb{E}_{q(\mathbf{x})}[\mathbf{f}(\mathbf{x})\hat{w}(\mathbf{x})] - \hat{\mathbf{J}}_N)^2 \right] \right)$$

$$\leqslant \frac{2}{(\mathbb{E}_{q(\mathbf{x})}[\hat{w}(\mathbf{x})])^2} \left( \mathbb{E}\left[ (\mathbb{E}_{q(\mathbf{x})}[\hat{w}(\mathbf{x})] - \hat{P}_N)^2 \right] + \mathbb{E}\left[ (\mathbb{E}_{q(\mathbf{x})}[\mathbf{f}(\mathbf{x})\hat{w}(\mathbf{x})] - \hat{\mathbf{J}}_N)^2 \right] \right)$$

$$= \frac{2}{(\mathbb{E}_{q(\mathbf{x})}[\hat{w}(\mathbf{x})])^2} \left( \text{Var}(\hat{P}_N) + \text{Var}(\hat{\mathbf{J}}_N) \right).$$

Since $\text{Var}(\hat{P}_N) = \frac{1}{N}\text{Var}(\hat{w}(\mathbf{x}))$, $\text{Var}(\hat{\mathbf{J}}_N) = \frac{1}{N}\text{Var}(\mathbf{f}(\mathbf{x})\hat{w}(\mathbf{x}))$, then we have

$$\mathbb{E}\|\hat{\mathbf{I}}_{\text{SNIS}} - \mathbf{I}\|^2 = \frac{2}{(\mathbb{E}_{q(\mathbf{x})}[\hat{w}(\mathbf{x})])^2 N} \left( \text{Var}(\hat{w}(\mathbf{x})) + \text{Var}(\mathbf{f}(\mathbf{x})\hat{w}(\mathbf{x})) \right)$$

$$\leqslant \frac{2}{(\mathbb{E}_{q(\mathbf{x})}[\hat{w}(\mathbf{x})])^2 N} \left( \mathbb{E}_{q(\mathbf{x})}[\hat{w}(\mathbf{x})^2] + \mathbb{E}_{q(\mathbf{x})}[\|\mathbf{f}(\mathbf{x})\hat{w}(\mathbf{x})\|^2] \right)$$

$$\leqslant \frac{4}{N} \frac{\mathbb{E}_{q(\mathbf{x})}[\hat{w}(\mathbf{x})^2]}{(\mathbb{E}_{q(\mathbf{x})}[\hat{w}(\mathbf{x})])^2},$$

where we use $\|\mathbf{f}\| \leqslant 1$ to get the last inequality. Hence, we have proved the first result. Similarly, we can prove the result for the bias. Since $\mathbb{E}\left[ \hat{\mathbf{J}}_N - \mathbb{E}_{q(\mathbf{x})}[\mathbf{f}(\mathbf{x})\hat{w}(\mathbf{x})] \right] = 0$, $\mathbb{E}\left[ \hat{P}_N - \mathbb{E}_{q(\mathbf{x})}[\hat{w}(\mathbf{x})] \right] = 0$, we have:

$$\|\mathbb{E}[(\hat{\mathbf{I}}_{\text{SNIS}} - \mathbf{I})]\| = \frac{1}{\mathbb{E}_{q(\mathbf{x})}[\hat{w}(\mathbf{x})]} \left\| \mathbb{E}\left[ \hat{\mathbf{I}}_{\text{SNIS}} \left( \mathbb{E}_{q(\mathbf{x})}[\hat{w}(\mathbf{x})] - \hat{P}_N \right) - \left( \mathbb{E}_{q(\mathbf{x})}[\mathbf{f}(\mathbf{x})\hat{w}(\mathbf{x})] - \hat{\mathbf{J}}_N \right) \right] \right\|$$

$$= \frac{1}{\mathbb{E}_{q(\mathbf{x})}[\hat{w}(\mathbf{x})]} \left\| \mathbb{E}\left[ \left( \hat{\mathbf{I}}_{\text{SNIS}} - \mathbf{I} \right) \left( \mathbb{E}_{q(\mathbf{x})}[\hat{w}(\mathbf{x})] - \hat{P}_N \right) \right] \right\|$$

$$\leqslant \frac{1}{\mathbb{E}_{q(\mathbf{x})}[\hat{w}(\mathbf{x})]} \left( \mathbb{E}\left[ \left\| \hat{\mathbf{I}}_{\text{SNIS}} - \mathbf{I} \right\|^2 \right] \right)^{\frac{1}{2}} \left( \mathbb{E}\left[ \left( \mathbb{E}_{q(\mathbf{x})}[\hat{w}(\mathbf{x})] - \hat{P}_N \right)^2 \right] \right)^{\frac{1}{2}}$$

$$\leqslant \frac{1}{\mathbb{E}_{q(\mathbf{x})}[\hat{w}(\mathbf{x})]} \left( \mathbb{E}\left[ \left\| \hat{\mathbf{I}}_{\text{SNIS}} - \mathbf{I} \right\|^2 \right] \right)^{\frac{1}{2}} \left( \frac{\mathbb{E}_{q(\mathbf{x})}[\hat{w}(\mathbf{x})^2]}{N} \right)^{\frac{1}{2}}.$$

By our first result, $\mathbb{E}\|\hat{\mathbf{I}}_{\text{SNIS}} - \mathbf{I}\|^2 \leqslant \frac{4}{N}\frac{\mathbb{E}_{q(\mathbf{x})}[\hat{w}(\mathbf{x})^2]}{(\mathbb{E}_{q(\mathbf{x})}[\hat{w}(\mathbf{x})])^2}$, then we have

$$\|\mathbb{E}[(\hat{\mathbf{I}}_{\text{SNIS}} - \mathbf{I})]\| \leqslant \frac{1}{\mathbb{E}_{q(\mathbf{x})}[\hat{w}(\mathbf{x})]} \left( \frac{4}{N}\frac{\mathbb{E}_{q(\mathbf{x})}[\hat{w}(\mathbf{x})^2]}{(\mathbb{E}_{q(\mathbf{x})}[\hat{w}(\mathbf{x})])^2} \right)^{\frac{1}{2}} \left( \frac{\mathbb{E}_{q(\mathbf{x})}[\hat{w}(\mathbf{x})^2]}{N} \right)^{\frac{1}{2}} = \frac{2}{N}\frac{\mathbb{E}_{q(\mathbf{x})}[\hat{w}(\mathbf{x})^2]}{(\mathbb{E}_{q(\mathbf{x})}[\hat{w}(\mathbf{x})])^2}.$$

So we have prove the second result. When $N \to \infty$, $\|\mathbb{E}[(\hat{\mathbf{I}}_{\text{SNIS}} - \mathbf{I})]\| \to 0$, then the SNIS estimator is asymptotically unbiased. $\qquad\square$

Assume the dataset $\{\mathbf{x}_0^{(i)}\}_{i=1}^N \overset{\text{i.i.d.}}{\sim} p_{\text{data}}(\mathbf{x})$. Then then estimator

$$\frac{\sum_{n\in[N]} \mathbf{x}_0^{(n)} K_t(\mathbf{x}_t, \mathbf{x}_0^{(n)})}{\sum_{n'\in[N]} K_t(\mathbf{x}_t, \mathbf{x}_0^{(n')})}$$

is the SNIS estimator of the posterior expectation $\mathbb{E}[\mathbf{x}_0 \mid \mathbf{x}_t]$. By Prop. 8, the SNIS estimator is asymptotically unbiased. Then the estimator is

$$\hat{B}_t = \frac{1}{M} \sum_{m\in[M]} \left\| \frac{\sum_{n\in[N]} \mathbf{x}_0^{(n)} K_t(\mathbf{x}_t^{(m)}, \mathbf{x}_0^{(n)})}{\sum_{n'\in[N]} K_t(\mathbf{x}_t^{(m)}, \mathbf{x}_0^{(n')})} \right\|^2$$

is also asymptotically unbiased. For more details on importance sampling, see (Robert et al., 1999).

# G  EXPERIMENTAL DETAILS

## G.1  OPTIMAL LOSS ESTIMATION

In this subsection, we show more experiments results of our cDOL estimator. Our cDOL estimator is concluded in Alg. 1.

---

**Algorithm 1** The corrected Diffusion Optimal Loss (cDOL) estimator

---

**input** Diffusion schedule $\alpha_t$ and $\sigma_t$, diffusion step $t$, training dataset $\{\mathbf{x}_0^{(n)}\}_{n\in[N]}$; number of repeats $R$, data sample $\mathbf{x}_0$ subset size $L$, $\mathbf{x}_t$ sample size $M$, correction parameter $C$.

**output** Estimation of the diffusion optimal loss $J_t^{(\mathbf{x}_0)^*}$ at $t$.

1: **for** $r \in [R]$ **do**
2:    Sample a data subset $\{\mathbf{x}_0^{(r,l)}\}_{l\in[L]}$ independently randomly from $\{\mathbf{x}_0^{(n)}\}_{n\in[N]}$;
3:    **for** $\tilde{m} \in [M]$ **do**
4:        Sample an index $l_{\tilde{m}}$ randomly from $[L]$;
5:        Construct $\mathbf{x}_t^{(r,\tilde{m})} = \alpha_t \mathbf{x}_0^{(r,l_{\tilde{m}})} + \sigma_t \boldsymbol{\epsilon}^{(\tilde{m})}$ with $\boldsymbol{\epsilon}^{(\tilde{m})} \sim \mathcal{N}(\mathbf{0}, \mathbf{I})$;
6:    **end for**
7:    Compute $\hat{B}_t^{\text{cDOL}^{(r)}}$ using Eq. (11) (where $K_t$ is defined in Eq. (8));
8: **end for**
9: Compute $\hat{B}_t^{\text{cDOL}} = \frac{1}{R}\sum_{r\in[R]} \hat{B}_t^{\text{cDOL}^{(r)}}$ and $\hat{A}$ using Eq. (7);
10: **Return** $\hat{A} - \hat{B}_t^{\text{cDOL}}$.

---

**Convergence of cDOL estimator.**  Our cDOL estimator has four parameters $(R, M, L, C)$. As mentioned in Sec. 3.4, we empirically choose $C = 4N/L$, the subset size L can be taken to fully utilize memory. The parameters $R, M$ should be large enough to ensure that the estimator converges. We perform a convergence analysis with respect to $R, M$ in the CIFAR-10 dataset to justify our choice of $M, R$. The results are summarized in Fig. G.1.1 and Fig. G.1.2. As shown in Fig. G.1.1, our cDOL estimator will converge when $R$ is large enough and can approximate the ground truth optimal loss accurately. Empirically, we find that $R = 3N/L$ is enough for an accurate estimate. Next, we verify that our cDOL estimator will converge when $M$ is large enough. As shown in Fig. G.1.2, we can see that the cDOL estimator converges when $M \approx 4L$.

**Efficiency.**  The computational complexity of the naive estimator $\hat{B}_t$ (Eq. (9)) is $\mathcal{O}(N^2)$, where $N$ is the size of the dataset. The cDOL estimator reduces this complexity to $\mathcal{O}(L^2 \times R)$. Based on the verifications the the previous paragraph, we set the subset size $L$ to fully utilize memory, and is often set to 2500 or 5000. We set $R = 3N/L$, $M = 4L$, then the total complexity becomes $\mathcal{O}(NL)$. With this setting, the total running time is approximately 0.5 hour on CIFAR-10, 2.5 hours on FFHQ, and about 1 day for the ImageNet dataset when using 2400 8G CPU cores.

## G.2  DIRECT COMPARISONS WITH WORKS ON OPTIMAL SOLUTION

Xu et al. (2023) proposes an SNIS estimator for the inner expectation of the optimal loss in Eq. (6). If $\mathbf{x}_t$ is sampled independently from a batch separate from the $\mathbf{x}_0$ batch used for the inner expectation, then the estimator reduces to the SNIS estimator described in Sec. 3.3. By contrast, if the same $\mathbf{x}_0$ batch is used to sample $\mathbf{x}_t$ and compute the outer expectation, this corresponds to our DOL estimator. As shown in Fig. 1, the SNIS estimator suffers from high variance, leading to poor empirical performance. Meanwhile, the DOL estimator introduces extra bias and also does not achieve good performance.

Niedoba et al. (2024) proposes a nearest neighbor estimator of the optimal solution. Given a noisy sample $\mathbf{x}_t$, the KNN estimator finds the $K$-nearest $\mathbf{x}_0$ samples in the dataset to estimate the optimal solution. The KNN search method used in (Niedoba et al., 2024) (Faiss with a flat index) has $\mathcal{O}(N)$ complexity per query of $\mathbf{x}_t$, leading to an overall complexity of $\mathcal{O}(N^2)$, which matches that of the naive estimator $\hat{B}_t$ (Eq. (9)). Moreover, KNN search requires significantly more memory, as it needs to generate an index of the entire dataset, which prevents effective multithreading parallelism.

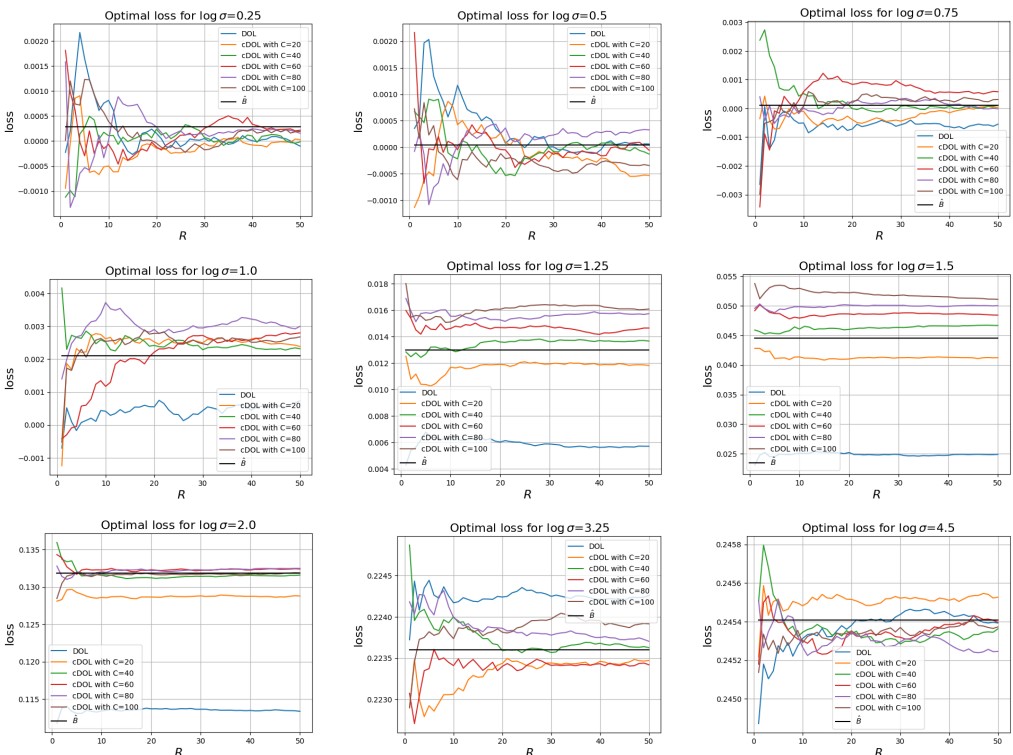

Figure G.1.1: Convergence of our estimator with respect to the number of subsets $R$ on CIFAR-10. We plot the estimated optimal loss v.s. $R$ for several choices of $C$ among different noise scales. For fair comparison, we fix the subset size $L = 5000$.

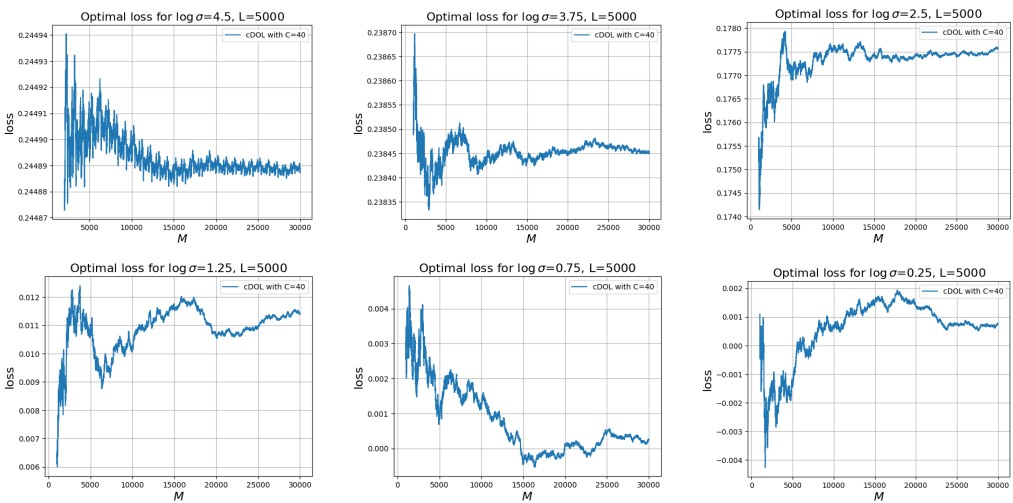

Figure G.1.2: Convergence of our estimator with respect to $M$ (the number of $\mathbf{x}_t$ samples) on CIFAR-10. We plot the estimated optimal loss v.s. $M$ for a fixed $C = 40, L = 5000$ among different noise scales.

For a direct comparison, we report the error rate of each estimator, defined as

$$e = \frac{|J_{\text{estimated}} - J_{\text{ground\_truth}}|}{J_{\text{ground\_truth}}}.$$

We consider the error rate more suitable than absolute error, since the scale of the optimal loss varies significantly across noise levels. For a fair comparison, we tested $L = 2500$ for the cDOL estimator and $n = K = 2500$ for the KNN estimator, the results are shown in Table G.2.1. We can see from the results that cDOL achieves comparable accuracy and variance to KNN, but with significantly lower runtime: approximately $5\times$ faster than KNN ($n = 2500, K = 2500$) due to its lower complexity $\mathcal{O}(L \times N)$ (with $R \propto N/L$) versus $\mathcal{O}(N^2)$ for KNN. Moreover, cDOL benefits from straightforward multithreading parallelism since it only loads $L$ samples into memory at a time, making it more scalable for large, high-resolution datasets such as ImageNet.

Table G.2.1: Comparison between different estimators on CIFAR-10 for an intermediate noise level $\log \sigma = 1.25$.

| Methods | DOL | cDOL ($L = 2500$) | KNN ($n = 2500, K = 2500$) | SNIS |
|---|---|---|---|---|
| error rate | 0.55 | 0.04 | 0.03 | 5.72 |
| variance (per dimension) | 0.0170 | 0.0182 | 0.0183 | 0.0210 |
| run time | 12 min | $\sim$12 min | $\sim$67 min | – |

## G.3 DETAILED SETTINGS AND ADDITIONAL RESULTS FOR FIG. 3

**Detailed settings.** Following EDM (Karras et al., 2022), we configure our training settings as follows. We train all models on CIFAR-10 until a total of 200 million images have been sampled from the training set. The batch size is set to 512. For sampling, we employ the EDM deterministic sampler, consistently setting the time steps according to $\sigma_i = \left(\sigma_{\max}^{1/\hat{\rho}} + \frac{i}{N-1}(\sigma_{\min}^{1/\hat{\rho}} - \sigma_{\max}^{1/\hat{\rho}})\right)^{\hat{\rho}}$, where $\hat{\rho} = 7, \sigma_{\min} = 0.002, \sigma_{\max} = 80$, and the number of function evaluations (NFE) is set to 35 for all models.

For training loss calculation, we evaluate the clean-data prediction loss for fair comparisons. In practice, we perform inference over three epochs to estimate the training loss across noise levels and observe that these estimates exhibit good convergence.

**Justification on loss estimation convergence.** To justify the convergence of training loss estimation, we present additional results using a model trained with the DDPM schedule (VP-$\epsilon$ in Table 1) on CIFAR-10. We computed the mean, variance, and standard error of the mean from 150,000 independent evaluations (corresponding to 3 epochs) of the training loss. The results (Table G.3.1) demonstrate that although the variance depends on the noise level, the largest standard deviation is still orders lower than the corresponding mean value. More accurate estimates can be obtained by increasing the number of model evaluations.

Table G.3.1: Training loss statistics across noise levels using a DDPM schedule on CIFAR-10.

| Noise level | Mean | Variance | Standard Error of Mean |
|---|---|---|---|
| $\log \sigma = 4$ | 0.230 | 0.015 | 3.95e-5 |
| $\log \sigma = 2$ | 0.130 | 0.0033 | 8.77e-6 |
| $\log \sigma = 0$ | 0.025 | 8.67e-5 | 2.23e-7 |
| $\log \sigma = -2$ | 0.0026 | 7.41e-7 | 1.91e-9 |

**Effect of sampling methods.** To decouple the possible influence of using different sampling methods, we make an investigation on the sampler effect. For a fair comparison, we use the EDM sampler with identical parameters to evaluate different training methods in Fig. 3. Additionally, we conducted experiments with the 250-step DDIM (Song et al., 2021a) and Flow Matching (Lipman et al., 2023) samplers, with the FID results presented in Table G.3.2. These results show that while sampling quality depends on the choice of sampler, a better-trained model consistently achieves higher sample quality across different samplers.

As we primarily evaluate the model by the FID metric in Fig. 3, we give some complementary results for Fig. 3 in this subsection.

Table G.3.2: FID ($\downarrow$) results across different samplers and training methods on CIFAR-10 with 250 sampling steps.

| Sampler\Training Method | EDM | DDPM | NCSN | FM | FM + **Our schedule** |
|---|---|---|---|---|---|
| EDM sampler | 1.94 | 1.97 | 2.72 | 2.36 | **1.79** |
| DDIM sampler | 2.14 | 2.23 | 2.91 | 2.27 | **1.99** |
| FM sampler | 2.19 | 2.25 | 3.07 | 2.28 | **2.04** |

**Precision and recall metrics.** As the FID metric is sensitive to both sample quality and diversity, it cannot reflect the sample diversity and quality separately. The precision and recall metrics are designed to test the sample quality and diversity, respectively. We evaluate our models trained by different training schedule and formulations in Fig. 3 under these two metrics. The results are shown in the following Table G.3.3.

Table G.3.3: Precision and recall results under different training schedules on CIFAR-10.

| Training schedule | Precision | Recall |
|---|---|---|
| EDM (Karras et al., 2022) | 0.615 | 0.682 |
| DDPM(Ho et al., 2020) | 0.608 | 0.683 |
| FM (Lipman et al., 2023) | 0.615 | 0.677 |
| SD3 (Esser et al., 2024) | 0.595 | **0.694** |
| NCSN (Song et al., 2021b) | 0.614 | 0.647 |
| **Our schedule** | **0.626** | 0.667 |

Combining the results in Fig. 3(a,b) with the results shown in the Table G.3.3, we observe that the precision metric also has a stronger correlation to the training loss gap in the small noise regions. Thus, our training schedule outperforms all other training schedules under this metric. In contrast, the recall metric has a stronger correlation to the training performance in the larger noise levels around the critical point $\sigma^\star$, thus the SD3 training schedule achieves the best performance. These results justify the intuition that the image quality is related to training performance at small noise scales, while recall or diversity is related to larger noise scales.

**Memorization metrics.** As shown in Fig. 3(a), the training loss gap is large for all mainstream diffusion models. This implies that these diffusion models are still not overfit to the optimal solution. To study the memorization behavior, we follow Gu et al. (2023) for the metric and the experimental settings. We train a model using Flow Matching precondition and our training schedule on a subset of CIFAR-10 with 5k data samples, and we train the same architecture with Flow Matching training schedule on the same dataset as a baseline. We report the memorization rate, where a sample is memorized if its $L^2$ distance to the nearest neighbor is smaller than $1/3$ of that to the second nearest neighbor in the training data (Gu et al., 2023). Here the factor $1/3$ is an empirical threshold proposed in Gu et al. (2023). The results are shown in the Table G.3.4. We can observe that our schedule improves the generation performance without leading to severe memorization.

Table G.3.4: Memorization rate ($\downarrow$) across different training epochs under different training schedules on CIFAR-10.

| Training schedule\Training Epochs | 0.5k | 1k | 1.5k | 2k | 2.5k | 3k | 3.5k | 4k |
|---|---|---|---|---|---|---|---|---|
| Flow Matching schedule | 0.0 | 0.0 | 0.0 | 0.0 | 0.0001 | 0.0024 | 0.0102 | 0.0224 |
| **Our schedule** | 0.0 | 0.0 | 0.0 | 0.0 | 0.0 | 0.0 | 0.0 | 0.0 |

## G.4 IMAGE GENERATION EXPERIMENTAL DETAILS

Following EDM (Karras et al., 2022), we configure our training settings as follows. We train all models on CIFAR-10 until a total of 200 million images have been sampled from the training set. The batch size is set to 512. Checkpoints are saved every 2.5 million images, and we report results

based on the checkpoint with the lowest FID. We adopt the DDPM++ network architecture used in EDM, with our primary modifications being the incorporation of our loss weighting scheme and adaptive noise distribution. All models are trained on 8 NVIDIA A100 GPUs. For sampling, we employ the EDM deterministic sampler, consistently setting the discretization steps according to $\sigma_i = (\sigma_{\max}^{1/\hat{\rho}} + \frac{i}{N-1}(\sigma_{\min}^{1/\hat{\rho}} - \sigma_{\max}^{1/\hat{\rho}}))^{\hat{\rho}}$, where $\hat{\rho} = 7, \sigma_{\min} = 0.002, \sigma_{\max} = 80$, and the number of function evaluations (NFE) is set to 35 for CIFAR-10 experiments.

For ImageNet-64, we follow a similar setup as EDM. We use the ADM architecture, which matches that of EDM (Karras et al., 2022). The batch size is set to 2048, and our loss weighting and adaptive noise distribution are applied as well. Training proceeds until 2.5 billion images have been sampled from the training set. Checkpoints are saved every 10 million images, and we report the checkpoint with the lowest FID. All ImageNet-64 models are trained on 32 NVIDIA A100 GPUs. In sampling, we again use the EDM deterministic sampler with $\hat{\rho} = 7$ and NFE = 79.

For ImageNet-256, we adopt a setup similar to LightningDiT (Yao et al., 2025). Specifically, we utilize VA-VAE (Yao et al., 2025) as the tokenizer and implement a modified LightningDiT (Yao et al., 2025) architecture enhanced with QK-Normalization (Dehghani et al., 2023) to improve training stability. The batch size is set to 2048, and we apply the same loss weighting and adaptive noise distribution strategies. Optimization is performed using AdamW with parameters $(\beta_1, \beta_2) = (0.9, 0.95)$ and a learning rate of $2 \times 10^{-4}$. The model is trained for 1600 epochs (approximately 10 million iterations), with checkpoints saved every 10,000 iterations. Again, we use 32 NVIDIA A100 GPUs to train the model on ImageNet-256. Consistent with LightningDiT (Yao et al., 2025), we employ the FM Euler ODE sampler with 250 function evaluations (NFE = 250).

**Details on our schedule.** We maintain a bin that records the training loss gap (note that the optimal loss is computed before training and does not need to be evaluated repeatedly; the loss gap can be evaluated only by monitoring the training loss) at each noise scale. Inspired by the adaptive schedule proposed by Kingma and Gao (2023), we update this bin using an exponential moving average (EMA) during training. Specifically, the bin is updated every time 2 million images have been drawn, with a decay rate set to 0.9. The collected training loss gap statistics are then used to construct a piecewise-linear probability density function, which serves as the adaptive noise schedule. As mentioned, our loss weight is given by Eq. (13):

$$w_\sigma = a \min\left\{1/J_\sigma^\star, w^\star\right\} + f(\sigma)\,\mathbb{I}_{\sigma<\sigma^\star}.$$

Typically, $w^\star$ is set to be 20 and $a = 1/50$. $f(\sigma) = \mathcal{N}(\log\sigma; \mu, \varsigma^2)$ is an additional weighting function to let us put more weight on the region $\sigma < \sigma^*$, which is simply set as a normal pdf. We set $\mu = -7.5, \varsigma = 2$ for CIFAR-10, $\mu = -5.75, \varsigma = 2$ for ImageNet-64 and $\mu = -4.37, \varsigma = 1.75$ for ImageNet-256.

In Fig. G.4.1, we plot our noise schedule calculated from the loss gap in the final optimization step of the training process. We can see from the result that our noise schedule indeed allocates more optimization steps on the region $\sigma < \sigma^\star$ with positive correlation to generation performance, as identified in Fig. 3. We also show some samples generated by our model trained on the ImageNet-256 dataset in Fig. G.4.2 and Fig. G.4.3.

**Human preference study.** We also conduct the human preference study to further evaluate the generative performance of different methods. We randomly generated 24 image pairs from the ImageNet-256 model trained from the baseline and our schedule. We asked 19 independent evaluators to select the image with better visual quality. The results are summarized in Table G.4.1 and show that our method was preferred in 55.26% of the cases, while the baseline was preferred in 44.73%. This human evaluation aligns with our FID improvements, confirming that the reduction in the "Loss Gap" translates into perceptibly better image quality.

Table G.4.1: Human preference between baseline and our schedule on ImageNet-256.

| Training schedule | FID with guidance (↓) | Human preference (↑) |
|---|---|---|
| LightningDiT (Yao et al., 2025) | 1.42 | 44.74% |
| **+ Our schedule** | **1.30** | **55.26%** |

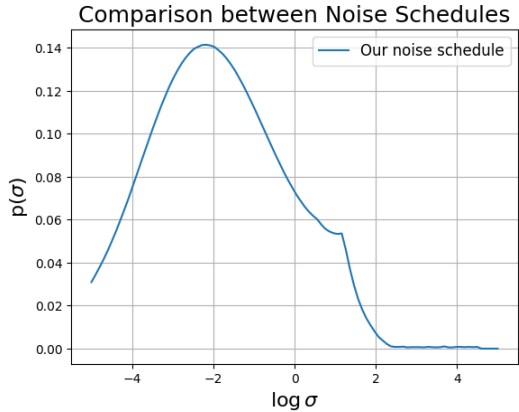

Figure G.4.1: Plot of our proposed noise schedule on the CIFAR-10 dataset. We plot the noise schedule calculated by the loss gap in the final optimization step of the training process.

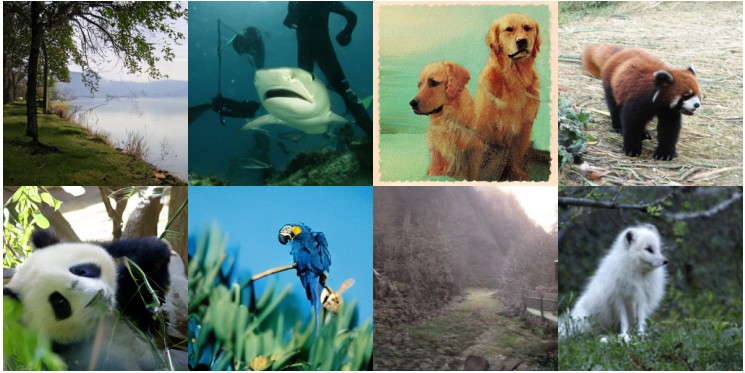

Figure G.4.2: Samples generated by our ImageNet-256 model.

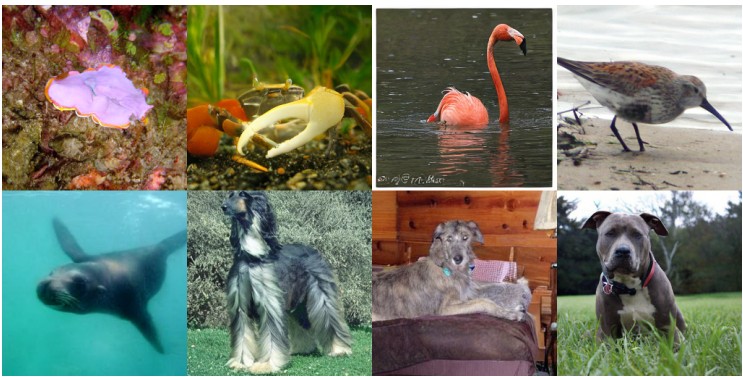

Figure G.4.3: Samples generated by our ImageNet-256 model.

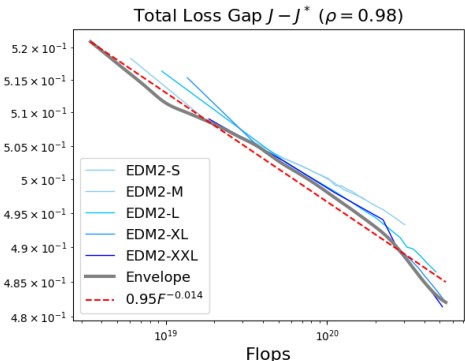

Figure G.4.4: Scaling law fitting results using the modified power law in Eq. (14) for the total diffusion loss on ImageNet-512.

## G.5 SCALING LAW EXPERIMENTAL DETAILS

In this subsection, we provide a comprehensive account of our scaling law analysis. Our experiments employ the state-of-the-art diffusion model EDM2 (Karras et al., 2024), with parameter counts ranging from 120M to 1.5B. In accordance with the training protocols outlined in EDM2, models are trained in the RGB space for ImageNet-64 and in the latent space derived from a pretrained VAE for ImageNet-512.

We begin by reporting the results on ImageNet-64. As described in Sec. 5, we apply our modified scaling law (Eq. (14)) to model performance. In Fig. G.5.1, we present a detailed comparison between the original and modified scaling laws across various noise scales. Our findings indicate that the modified formulation yields a loss envelope that adheres more closely to a linear relationship, with corresponding improvements in the correlation coefficient, especially at large noise scales. The enhancements at small noise scales are less pronounced, largely due to the relatively minor optimal loss values compared to the actual training losses.

Subsequently, we extend our analysis to ImageNet-512. Mirroring the experimental setup used for ImageNet-64, we adopt the optimized adaptive loss weighting from EDM2 (Karras et al., 2024) when calculating total loss. The results, shown in Fig. G.4.4, achieve a correlation coefficient of $\rho = 0.9857$, with the fitted scaling law given by:

$$J(F) = 0.9493F^{-0.014} + 0.001.$$

A thorough comparison between the original and modified scaling laws across multiple noise scales is presented in Fig. G.5.2. These results are consistent with those observed on ImageNet-64, once again demonstrating that the modified scaling law yields a loss envelope that is closer to linearity, with higher correlation coefficients, particularly at large noise scales. As before, improvements at small noise scales remain limited due to the diminutive size of optimal loss relative to training loss.

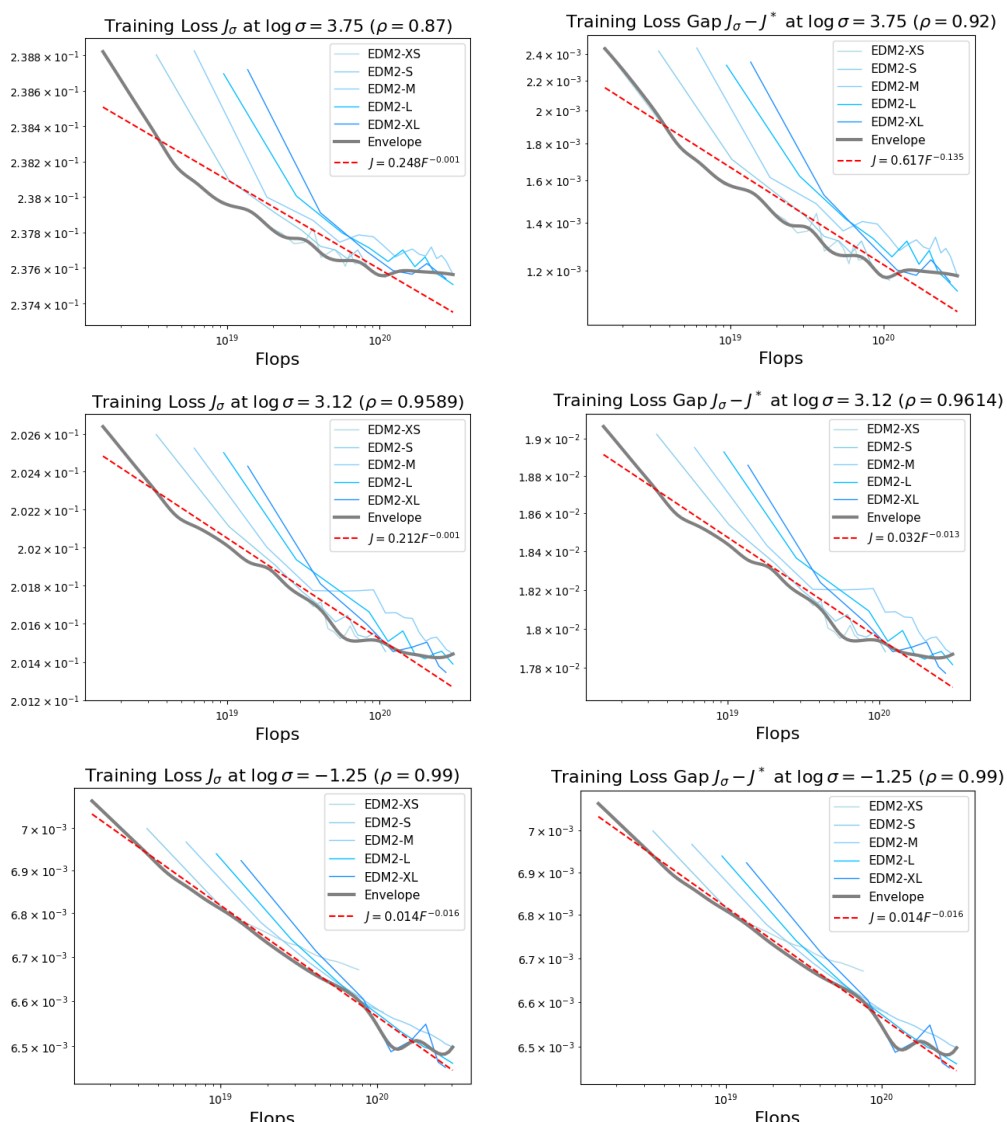

Figure G.5.1: Scaling law study on ImageNet-64. Each row corresponds to a different noise scale. The left column shows the raw training loss values, while the right column displays the training loss gap relative to the optimal loss at each noise scale. We observe that in the modified version, the envelope aligns more closely with a straight line, particularly at larger noise scales. For smaller noise scales, the improvement is less pronounced, since the optimal loss is still very small compared to the training loss of the model.

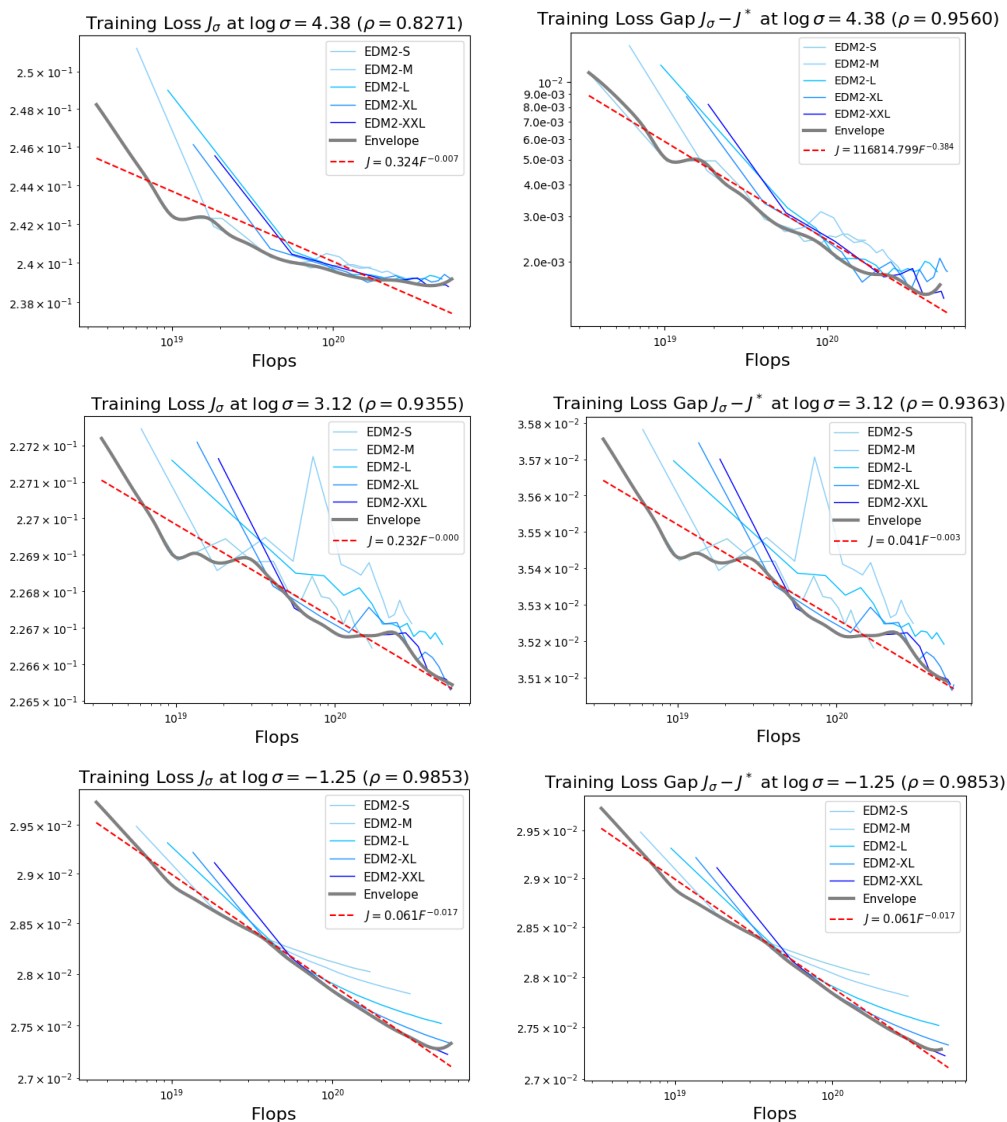

Figure G.5.2: Scaling law study on ImageNet-512. Each row corresponds to a different noise scale. The left column shows the raw training loss values, while the right column presents the training loss gap relative to the optimal loss at each noise scale. We observe that in the modified version, the envelope aligns more closely with a linear trend, especially for larger noise scales. For smaller noise scales, the improvement is less significant, as the optimal loss is still very small compared to the training loss of the model.

