# OpenReview forum: "Diagnosing and Improving Diffusion Models by Estimating the Optimal Loss Value"
_ICLR.cc/2026/Conference — ICLR 2026 Poster_

### Official Review · Reviewer_Bp46 · 2025-10-27

**Soundness:** 3
**Presentation:** 3
**Contribution:** 3
**Rating:** 6
**Confidence:** 3

**Summary:**

The paper targets a known limitation of diffusion training losses—namely, that the theoretical optimum is generally non-zero and unknown—making the raw loss an unreliable proxy for data-fitting quality. The authors (i) derive a closed-form expression for the optimal loss under a unified formulation of diffusion models, (ii) propose practical estimators, including a stochastic variant with variance/bias control scalable to large datasets, and (iii) use these tools to diagnose training quality and design an improved training schedule.

**Strengths:**

Theoretical derivations appear rigorous.
Writing quality and notation are clear, consistent, and professionally presented.

**Weaknesses:**

1. Loss–performance alignment under matched settings. The paper argues it resolves the mismatch between diffusion loss (with unknown non-zero optimum) and generative performance. To substantiate this claim more convincingly, I suggest including a correlation analysis between training loss (after subtracting the optimal term) and FID under identical settings and budgets, compared to standard loss formulations. Concretely, report Pearson/Spearman correlations over training, rank consistency across checkpoints, and side-by-side training-curve plots for your method vs. baselines on the same architectures/datasets.

2. Over-reliance on FID. FID alone cannot fully capture generative performance. Consider adding complementary metrics and qualitative evidence. A small human preference study (even limited-scale) would help validate perceptual quality beyond distributional statistics.

**Questions:**

See the weaknesses.

---

> ### Author Response · Authors · 2025-11-27
> **Rebuttal**
>
> Thank you for the recognition of our contributions. We also appreciate your valuable questions and suggestions, which help us further improve our paper. Here are our responses to your questions:
>
> ## About the loss-performance alignment.
> Thank you for bringing up this informative question. We would like to first mention that we are __not__ claiming that there is a mismatch between the actual (i.e., the conventional / standard) diffusion loss (summed over all diffusion noise scales (or, time steps) under a given loss weight) and generation performance; in fact, as the actual loss and the loss gap (the result by subtracting the optimal loss value from the actual training loss) differ only by a constant (the optimal loss value only depends on the training dataset), so under identical settings and budgets, the Pearson/Spearman correlation between the actual loss and FID, and the Pearson/Spearman correlation between the loss gap and FID, are identical (the Pearson/Spearman correlation is invariant if one variable is shifted by a constant).
>
> Indeed, following your instruction, we plot the loss gap vs. FID and compare with the plot of training loss vs. FID along the training process on the CIFAR-10 dataset under the Flow Matching formulation using the EDM's model architecture. The Pearson/Spearman correlations are calculated for the two cases. The results are shown in Fig. 8 in Appx.H.5. The correlations are indeed identical in the two cases.
>
> In contrast, the claim in our paper is that the loss gap measures the __absolute__ fit to the optimal solution (i.e., the learning target) at each diffusion noise scale (or, time step), which identifies the noise-scale region that is more critical to generation performance (and leads to more principled scaling law study) hence instructs the design of a principled training schedule.
> More concretely, in Fig. 3(a) and (b), we found that the noise-scale-wise loss gap (as well as the noise-scale-wise actual diffusion loss, as explained above) after training convergence has qualitatively different correlations (i.e., ranking order) to the FID in different noise-scale regions (due to the trade-off among optimizing the loss at different noise scales; see also the answer to Reviewer yn5v, Question 6). The separation between the positively correlated noise-scale region and negatively correlated noise-scale region is identified by $\sigma^\star$, the point where the optimal loss value first becomes close to zero. A principled training schedule can then be designed, which puts larger weights on the positively correlated noise-scale region, whose location is informed by $\sigma^\star$ from the optimal-loss analysis.
>
> ## Complementary metrics and qualitative evidence
> Thanks for the suggestion. We are conducting a human preference study to further evaluate the generative performance of different approaches. We still need few days to complete the evaluation, and we will update the results to you once the evaluation is completed. Besides the human preference study, we also test the Inception core, Precision, and Recall metrics of different methods. The results are summarized in Table 7 in Appx.H.4 and Table 3 in the main text.
>
> ---
> Thank you again for your detailed suggestions and insightful questions, which help to improve the quality of our text. If you have any further questions and suggestions, we are glad to discuss with you.

---

> > ### Author Response · Authors · 2025-12-03
> > **Update of Rebuttal**
> >
> > We have completed the human preference study to further evaluate the generative performance of different approaches, and glad to update the results with you.
> >
> > - Setup: We randomly generated 24 image pairs (Baseline vs. Ours) from the ImageNet-256 model. 19 independent evaluators were asked to select the image with better visual quality.
> > - Results: The results (summarized in Table 9) show that our method was preferred in 55.26\% of the cases, while the baseline (LightningDiT) was preferred in 44.73\%.
> > - Conclusion: This human evaluation aligns with our FID improvements, confirming that the reduction in the "Loss Gap" translates into perceptibly better image quality.
> >
> > We believe these multifaceted evaluations provide robust evidence of our method's effectiveness.

---

### Official Review · Reviewer_yn5v · 2025-10-28

**Soundness:** 3
**Presentation:** 3
**Contribution:** 3
**Rating:** 8
**Confidence:** 3

**Summary:**

This paper introduce the optimal loss in the parameter designs of diffussion models. The authors propose an estimator of optimal loss. The loss gap idea is then used in the design of noise schedules and loss weights in training objectives. The scaling law of diffusion model is also discussed based on the loss gap other than just the loss value.

**Strengths:**

The authors propose the use of optimal loss in training schedule design, and design practical estimation of the optimal loss. They also proposed a new schedule which seems to be some refined version of EDM schedules but focuses more about the nature of loss gap between the true loss and the optimal loss, and the practical performance of their schedule is validated.

**Weaknesses:**

The proposed schedule is still vague, both in writing and in the actual design (see Questions).

**Questions:**

- In section 2, sometimes the data distribution is $p_0(x_0)$, sometimes is $p(x_0)$. Are they different distributions?
- Please specify the noise schedule $p(t)$ and weight $w_t$ in line 293 using the notation from Section 2 (especially which superscript should be used).
- In Line 382, please specify what is v, F prediction.
- In Line 386-387, is the loss weight discontinuous, which contradicts the continuity of actual loss gap? If so, then why this is a good choice?
- In Line 396, why $p(\sigma)$ is chosen to be proportional to $w_{\sigma}(J-J*)$ instead of for example $(w_{\sigma} (J-J^*))^a$ where $a\ge 0$ ? Is there a theoretical/empirical reason for this? What precise noice schedule do you use in the experiments? Is this changing at each training iteration due to different $J_\sigma(\theta)$? If there is no explicit expression, please show some figures of $p(\sigma)$.
- The strategy of choosing weights and noise schedule is quite confusing to me. In loss weight, you would like to emphasize left side of the threshold $\sigma*$. Btw, the authors state in the paper but I'm still confused why such pattern (the emphasis of positive correlation) is desired. In noise schedule, you would like to emphasize  $w_\sigma (J-J*)$ and therefore in total, combining loss weight, noise schedule, and the l2 distance, the loss looks something like $w_\sigma^2 (J-J*) J^2$ (or maybe $w_\sigma^2 (J-J*) (J^2-J^{*2})$ since constant term does not affect the optimization). Can the author explain the overall intuition of the schedule?
- What is the schedule (timestep, discretization) of $\sigma$? Is this also adapted in some way?


Minor:
- font size in figures is too small

---

> ### Author Response · Authors · 2025-11-27
> **Rebuttal (Part 1/2)**
>
> Thank you for your detailed review and for characterizing our contribution from both empirical and theoretical aspects. We also appreciate your valuable suggestions and questions, which help us improve our paper. Here are our responses to your every question, which we hope to help you better understand our proposed schedule:
>
> ## Different notations of the data distribution ($p_0(x_0)$ and $p(x_0)$) (Question 1)
> Thanks for your careful reading. They are indeed typos, and both of these notations are used to denote the data distribution. We have fixed the typos in the revised version of our paper.
>
> ## Specifying the noise schedule $p(t)$ and weight $w_t$ in line 307 (Question 2)
> Thank you for this point. We would like to first clarify that the sentence in between lines 307-308 is to define the "training schedule" term, which collectively includes both the noise schedule and loss weight concepts. Here, we do not step into any specific approaches yet, so we do not ground these two notations into any specific designs.
>
> In the following Section 4.1, we begin to analyze different training schedules ($p(t),w_t$) through the optimal loss perspective, which builds upon the equivalent conversion among different diffusion formulations. This enables us to use the clean-data prediction ($\mathbf{x}_0$-prediction) for unification (line 337-353). Therefore, in the following contents we specify $p(t)$ and $w_t$ using the $\mathbf{x}_0$-prediction notation from Section 2 (Eqn.(4)). We have correspondingly revised our paper and hope it can help the readers clarify these notations.
>
> ## Specifying what is v, F prediction (Question 3)
> Thanks for mentioning this line. The "$\mathbf{v},\mathbf{F}$ prediction" refers to $\mathbf{v}$-prediction and $\mathbf{F}$-prediction. The $\mathbf{v}$-prediction is short for vector-field prediction (line 108-111), and the $\mathbf{F}$-prediction corresponds to the EDM's prediction formulation (Table 1, VE-$\mathbf{F}$ and Line 353). We have revised our paper to explicitly define these abbreviations right after their full names/definitions.
>
> ## Is the loss weight discontinuous? (Question 4)
> Thank you for your insightful question. In fact, our designed loss weight is continuous in practice, which aligns with the continuity of the actual loss gap. As mentioned in our paper, our choice of weight is given by $w_\sigma = a \min(\frac{1}{J_\sigma^\star},w^\star) + f(\sigma) I_{\sigma < \sigma^\star}$, where $f(\sigma) = \mathcal{N} (\log \sigma; \mu, \varsigma^2)$.
> In practice, we choose the hyperparameters $\mu, \varsigma$ such that $\sigma^\star>\mu+3\varsigma$ (details in Appx.H.5), thus we find that $f(\sigma)$ has very small value on the region $\sigma>\sigma^\star$, so we just use $w_\sigma = a \min(\frac{1}{J_\sigma^\star},w^\star) + f(\sigma)$ in practice.
>
> ## Why $p(\sigma)$ is chosen to be proportional to $w_\sigma(J-J_\star)$? (Question 5)
> Thanks for this insightful question. We choose $w_\sigma(J-J_\star)$ instead of $(w_\sigma(J-J_\star))^a,a>0$ just out of simplicity. As we claimed in the paper, we believe a desired $p(\sigma)$ should favor noise steps on which the optimization task has not yet been done well. That is to say, for each optimization step, we would like to make the update in similar scales. The detailed formulation of $p(\sigma)$ is. We also conduct further experiments following your idea that $p(\sigma)\propto(w_\sigma(J-J_\star))^a,a>0$ on the CIFAR-10 dataset. All the models are trained with the same architecture as EDM and using our loss weight. The results are shown in the following table.
>
> |Settings|FID|
> |-|-|
> |$a=1.2$|1.81|
> |$a=0.8$|1.80|
> |$a=1.0$ (Our choice)| 1.79|
>
> From the results, we observe that the generation performance is not sensitive to the hyperparameter $a$. So a simple setting of $a=1$ is enough for improving the generation performance. Regarding our adaptive training schedule, we maintain a bin that records the training loss gap for
> each noise scale. Inspired by the adaptive schedule proposed by [1], we update
> this bin using an exponential moving average (EMA) during training (details in Appx.H.5). So our schedule is indeed changing with the training loss. We plot our noise schedule $p(\sigma)$ calculated by the model’s loss gap in the final optimization step of the training process on CIFAR-10 in Fig. 7, Appx.H.5. We can see from the result that our model allocates
> more optimization steps on the positive region, as shown in Fig. 3.
>
> [1]. Diederik P Kingma and Ruiqi Gao. Understanding diffusion objectives as the ELBO with simple
> data augmentation. NeurIPS 2023.

---

> > ### Author Response · Authors · 2025-11-27
> > **Rebuttal (Part 2/2)**
> >
> > ## The overall intuition of the schedule.  (Question 6)
> > Good catch. Here we would like to respond to your question in detail regarding both our loss weight and noise schedule:
> > (1) Why the emphasis on positive correlation desired for our loss weight: good catch. Our design choice is motivated by our **Loss gap vs. generation performance** analysis (the last paragraph in Section 4.1). In detail, we analyze how the loss gap can be used as data-fitting measure to analyze **which region of different noise levels** is more critical for the generation performance (Fig.3). After comparing different diffusion models, we obtain several observations:
> > - At large $\sigma$, an erroneous fit leads to a deficiency in generation quality;
> > - If a good fit at large $\sigma$ is achieved, $\operatorname{log}\sigma\in[-2.0,2.0]$ becomes more relevant;
> > - For $\operatorname{log}\sigma\in[-2.0,2.0]$, the correlation between loss gap and FID becomes positive only in the region further left to $\sigma^\star$ (the largest $\sigma$ whose optimal loss $J_{\sigma}^*$ becomes positive).
> >
> > Combining the above analysis,  we choose to use precondition coefficients $c_\sigma$ proposed by VE-F(EDM) and FM-v(FM) (in Table 2) to ensure a good fit in large $\sigma$ region and design the training schedule to put more training efforts on the positive region (L373-377).
> >
> > (2) The understanding of our loss and training schedule.
> > We want to clarify that our training loss is not $w^2_\sigma (J_{\sigma}-J_{\sigma}^\star)J_{\sigma}$, because we does not perform gradient backpropagation when calculating the noise scale $\propto w_{\sigma}^2(J_{\sigma}-J_{\sigma}^\star)$. Theoretically, the loss weight and the noise schedule are the same as they are both coefficients multiplied by the training loss. However, these two components serve as different roles in practice. As we have mentioned in L402-404 and L412-414, $w_\sigma$ calibrates the error resolution across different noise scales and $p(\sigma)$ allocates the optimization frequency to each noise level. The intuition of our schedule is that the "importance" of the noise scales is encoded in the loss weight $w_\sigma$, and $p(\sigma)\propto w_{\sigma}^2(J_{\sigma}-J_{\sigma}^\star)$ allocates optimization steps during the training process according to the training loss gap of the current model.
> >
> > ## The schedule (timestep, discretization) of $\sigma$.  (Question 7)
> > In this work, we focus on the training behaviors of different diffusion model formulations. We keep the sampler fixed so the schedule of $\sigma$ is not adapted. In Table 6 we also present results across different samplers and training methods. For the CIFAR-10 and ImageNet-64 experiments, we always set $\sigma=t$ and use the EDM discretization, which is the same as the choice of the EDM baseline. For the ImageNet-256 experiments, we use the same sampler as the baseline LightningDiT, which is a Flow Matching sampler with $t=\frac{\sigma}{1+\sigma}$ and Stable Diffusion 3 discretization. The detailed settings are reported in Appx.H.5.
> >
> > ## The font size in figures.  (Question 8)
> > Thank you for the suggestions. We have reorganized and revised the figures to improve the visualization.
> >
> > ---
> > Thank you again for your detailed suggestions and insightful questions, which help to improve the quality of our text. If you have any further questions and suggestions, we are glad to discuss with you.

---

### Official Review · Reviewer_LMDA · 2025-10-30

**Soundness:** 3
**Presentation:** 2
**Contribution:** 3
**Rating:** 6
**Confidence:** 3

**Summary:**

The paper investigates how to estimate the optimal loss values in diffusion models’ training objectives. It first introduces several diffusion losses (corresponding to different prediction targets) and presents them under a unified formalism and notation. Since the optimal predictors for these objectives are given by conditional expectations, their minimal achievable losses still depend on the conditional variances. Based on this, Theorem 1 derives an analytical expression for the optimal loss of the clean-data prediction objective, showing that it decomposes into two terms: a simple dataset-average term and a second, computationally expensive nested expectation. One of the paper’s main contributions is to propose efficient approximations for this second term—the DOL and cDOL estimators—which make estimation feasible on large datasets. The authors validate these estimators on smaller datasets, demonstrating that the approximated optimal losses closely match their true values across diffusion times. They then use the estimated optimal losses to analyze how existing diffusion models approach these limits as noise varies, revealing distinct behaviors across diffusion steps. These observations motivate a combination of an improved training schedule and a loss-weighting scheme yielding better FID scores. Finally, the paper revisits scaling laws for diffusion models, showing that when losses are shifted by their corresponding optimal values, the resulting curves adhere more closely to power-law behavior.

**Strengths:**

- From the best of the reviewer's knowledge, the characterisation of the diffusion optimal loss values is novel.
- The paper's results are quite general as the manuscript is structured in such a way that several diffusion setups (i.e., different prediction targets and noising schemes) can be analysed under the same framework.
- The use of optimal loss values as a diagnostic tool is interesting as it allows to identify the noise regimes where models are weaker and ultimately provide adjustment with improved nosing and loss-weighting schemes.

**Weaknesses:**

- Clarity of exposition and results presentation could be enhanced. The notation is quite heavy and the presentation of the results would benefit from more explanations/interpretation (see Questions). For example, it took me I while to understand what Figure 2(b) is plotting.
- More fundamentally, it is not clear whether getting a smaller training loss value ultimately results in better performing/generalising models. For example, it would be interesting to understand to what extent the proposed scheduler and loss weight impact memorisation.

**Questions:**

- Do the authors have an intuition behind the larger loss gap in the intermediate region? This gap appears to be consistent across various model variants.
- Figure 1(c): in the paper the authors state: "The result confirms that the variance increases with C." This does not seem apparent in the plot. could the authors please clarify?
- Could the authors please clarify what is on the y-axis in Figure 2(b)?
- From my understanding, the proposed loss weight tries to assign more importance to the interval to the left of $\sigma*$ based on the results in Figure 2(b). This is because the loss-gap positively correlates with FID in that region. However, the same figure also shows a negative correlation in the proximity of $\sigma*$. Does the proposed loss weight take that into account?
- Figure 3(b): why the choice of $\sigma = 4.38$?
- Figure 3(c): what do you mean by total loss gap?

---

> ### Author Response · Authors · 2025-11-27
> **Rebuttal (Part 1/2)**
>
> Thank you for your detailed review and believe our work is novel. We also appreciate your valuable suggestions and questions, which help us improve our work. Here are our responses to your questions.
>
> ## About the presentation of our work (Weakness 1 and Question 3)
> Thank you for your valuable feedback, and we apologize for missing the possible understanding difficulty of the current description. We have revised and updated our paper following your questions and suggestions.
> Particularly, rather than described as the "relative training loss gap", the y-axis in Fig. 3(b) would be better described as the "normalized training loss gap", where the normalization means dividing the loss-gap value on each curve at each diffusion time step by the average loss-gap value at that time step averaged over the four mainstream diffusion-model choices in the figure (the "VE-$\epsilon$" curve is excluded as it manifests a salient deviation from other curves and its FID is also obviously worse than others). It is for a clearer comparison/ranking over the four approaches at each diffusion time step.
> We have revised the description for Fig. 3(b) as well as Fig. 3(c) in the revised paper (caption of Fig. 3, marked blue).
>
> ## The memorization study (Weakness 2)
> Thank you for your insightful question. As shown in Fig. 3(a), although different approaches manifest a difference in the converged training loss, they all still leave a certain level of loss gap from the optimal loss values, especially for intermediate noise scales (explained in the next piece of response). This implies that it is not easy to overfit to the optimal solution in contemporary diffusion models.
> Particularly, Fig. 3(b) and (c) show that different training schedules, including ours, show a trade-off between the loss values at different time steps, i.e., there is not a single schedule that achieves a lower loss value over all the time steps. In this sense, we are not trying to lower the training loss but to identify a better trade-off for learning on different time steps. As for the estimation of the optimal loss, it is only used for endowing the training loss with the utility for measuring the absolute fit to training data henceforth for analyzing and improving diffusion model training behaviors. This does **not** indicate that we are driving the training loss as close to the optimal loss (perfect fit to training data) as possible.
>
> We shall show that our scheduler even leads to a less memorization rate. To study the memorization behavior, we follow [1] for the metric and the experimental settings. We choose a setting that adopts the "FM-v" formulation on a subset of CIFAR-10 with 5k data samples, and train a model using our training schedule compared with a model trained using the Flow Matching (FM) schedule as a baseline. We report the memorization rate, where a sample is regarded as memorized if its $L^2$ distance to the nearest neighbor is smaller than 1/3 of that to the second nearest neighbor in the training data [1]. Here the factor 1/3 is an empirical threshold proposed in [1]. The results are shown in the following table. We can observe that our schedule improves the generation performance without leading to severe memorization. We have included these results in our revised paper (Appx.H.4).
>
> |Training schedule\ Training Epochs|0.5k|1k|1.5k|2k|2.5k|3k|3.5k|4k|
> |-|-|-|-|-|-|-|-|-|
> |FM-v (FM) |0.0|0.0|0.0|0.0|0.0001|0.0024|0.0102|0.0224|
> |FM-v (ours) |0.0|0.0|0.0|0.0|0.0|0.0|0.0|0.0|
>
> [1] Gu, Xiangming, et al. "On Memorization in Diffusion Models." Transactions on Machine Learning Research.
>
> ## Explanation of the larger loss gap in the intermediate noise levels (Question 1)
> Thank you for your insightful question. We believe that it is a reasonable phenomenon because the diffusion model learns a trivial solution in extremely large and small noise levels. To see this, we consider the $x_0$ -prediction without loss of generality. When the noise level $\sigma$ is large, $x_\sigma=x_0+\sigma\epsilon$ is dominated by the noise term. Thus the optimal solution of the diffusion model becomes $E[x_0|x_\sigma]\approx E[x_0]$. Usually, the image data is normalized and $E[x_0]\approx 0$. Thus, the neural network only needs to simply output a constant 0 in large noise level region. For an extremely small noise level, the optimal diffusion model $E[x_0|x_\sigma]\approx x_\sigma$, i.e., the model tends to learn an identity map. Thus, the diffusion model tends to have a trivial solution in extremely large and small noise levels. The learning task is more difficult at intermediate noise levels, which leads to a larger loss gap.

---

> > ### Author Response · Authors · 2025-11-27
> > **Rebuttal (Part 2/2)**
> >
> > ## The variance of the cDOL estimator (Question 2)
> > Thank you for your question. We can see from Fig. 2 that when $C=\infty$(SNIS), the estimator has a large variance that leads to poor performance. This happens especially in the intermediate and small noise levels (we choose an intermediate noise level $\log\sigma=1.25$ in Fig.2, because in this case the variance is dominated by the term $\mathbf{x_0^{(l_\tilde{m})}}K(\mathbf{x_t^{(\tilde{m})}},\mathbf{x_0^{(l_\tilde{m})}})$ in Eqn.(11). To see this, when the noise level $\sigma_t$ is small ($t\to0$), the denominator of Eqn.(9) is dominated by $K_t$ (defined in Eqn.(8)) calculated by $(\mathbf{x}_t, \mathbf{x}_0)$ pairs which has the smallest $L^2$ distance. As the noise level is small, the $(\mathbf{x}_t, \mathbf{x}_0)$ pairs where $\mathbf{x}_t$ is constructed from $\mathbf{x}_0$ become the dominant term. Thus, if such a term is reduced ($C=\infty$, SNIS case), the estimation of Eqn.(11) becomes inaccurate, which is reflected by large variance. When $C$ is not $\infty$ even if it's a large constant, this term still becomes the dominant term in the small noise level region, thus helping to reduce variance and improve the accuracy. This is the reason why we do not observe a significant variance growth as $C$ increases.
> >
> > ## About our training schedule (Question 4)
> > Thank you for your question and detailed observation. As you suggested, we take the position of the positive interval into consideration when designing the training schedule. We show our noise schedule $p(\sigma)$ on the CIFAR-10 dataset in the revised paper (Fig.7). We plot the noise schedule
> > calculated by the model’s loss gap in the final optimization step of the training process. From Fig. 7, we can see that our schedule indeed puts more optimization steps on the positive region identified in Fig.3(b).
> >
> > ## Explanation of Fig. 4(b) (Question 5)
> > Thank you for your question. We want to clarify that we conduct the scaling law study on a uniform grid of noise levels in log scale. We report the representative results at a large noise scale $\log \sigma=4.38$ in the main text and give the comprehensive results in Appx.H.6. We choose the noise level $\log\sigma=4.38$ because the largest noise scale in our sampling process is $\sigma_{max}=80$, which is the choice of the EDM sampler. Since $\log \sigma_{max}=\log 80\approx4.38$, this noise level can be viewed as the largest noise scale that contributes to our generation process. The improvement of our modified scaling law is more pronounced for large noise scales, since the optimal loss is close to the model’s training loss in this region.
> >
> > ## The term "total loss gap" (Question 6)
> > The term total loss gap is defined by the training loss gap multiplied by the loss weight under the expectation of the noise schedule. In other words, this term refers to the real training loss in the training process minus the corresponding optimal value. We have added extra explanation on this term in the revised paper to improve the clarity of our presentation (in the caption of Fig.4, marked blue).
> >
> > ---
> > Thank you again for your detailed suggestions and insightful questions, which help to improve the quality of our text. If you have any further questions and suggestions, we are glad to discuss with you.

---

### Official Review · Reviewer_AWVe · 2025-10-31

**Soundness:** 4
**Presentation:** 2
**Contribution:** 4
**Rating:** 8
**Confidence:** 4

**Summary:**

This paper proposes to estimate the *optimal loss value* of diffusion models — a quantity that depends solely on the dataset and the diffusion setting, not on model capacity. The authors provide a theoretical derivation of this optimal loss under a unified formulation of diffusion models, and develop consistent and scalable estimators that remain stable and “not too greedy.” These estimators allow a principled analysis of model training quality, identifying where diffusion models underfit across noise scales. Leveraging these insights, the authors propose a new training schedule that equalizes performance across diffusion scales and leads to consistent improvements across various datasets (CIFAR-10, ImageNet-64, ImageNet-256). Finally, they show that subtracting the optimal loss from the observed training loss yields a more faithful scaling law, providing a better theoretical understanding of diffusion model scaling.

**Strengths:**

- The paper answers and analyzes questions that have been so far addressed only empirically — *finally!* The EDM sampling schedule has performed remarkably well for years, and this work explains *why*. Such principled analysis is much needed in a community increasingly driven by empirical results rather than theoretical foundations.
- Strong and complete theoretical analysis of the proposed estimators, including bias–variance tradeoffs and formal proofs of consistency.
- Extensive and well-structured experimental validation, showing clear and consistent improvements across multiple datasets and model variants.

**Weaknesses:**

- The presentation quality is poor. Example given: Figure 1 is hardly readable without zooming, and inline equations significantly hurt readability. Inconsistent line heights (e.g., lines 130–131, or the paragraph 189–200) make the paper visually dense and tiring to read.
- The alternative formulation of diffusion models in the main text feels unnecessary. A single sentence linking Theorem 1 to the alternative formulation in the appendix would suffice, leaving more space to improve layout and readability.
- While technically sound, the paper’s exposition could be more accessible, especially in highlighting intuition behind the estimators.

**Questions:**

- Intuitively, image *quality* tends to be related to predictions at small noise scales, while *recall/diversity* corresponds to large noise scales. It would be interesting to compare the loss gap across these regimes with standard Precision and Recall metrics — this could provide an even more interpretable connection between the optimal loss and generative performance.

**Details Of Ethics Concerns:**

--

---

> ### Author Response · Authors · 2025-11-27
> **Rebuttal**
>
> Thank you for your detailed review and your acknowledgment on our contributions from both the theoretical and empirical sides! We feel honored to know that you believe our efforts are meaningful for the community. We also appreciate your valuable suggestions and questions, which help us further improve our paper. Here are our responses to your questions.
>
> ## Presentation of the paper.
> Thank you for your valuable feedback. We understand your feeling of visual density due to the limitation of the page. Following your guidelines, we have uploaded our revised paper by (1) adjusting Fig. 1; (2) aligning line heights; (3) polishing the equations for better readability. Any further suggestion and feedback on our revised paper is appreciated.
>
> ## Alternative formulations of the diffusion model
> Thank you for your feedback. This is a helpful suggestion that helps to improve the readability of our paper. However, according to Questions 2 and 3 raised by Reviewer yn5v, we find that some discussions on the alternative formulation of the diffusion model are perhaps necessary for the presentation of Sec.4. So following your suggestions, we have simplified this section and moved the details of the derivation to Appx.C. Thank you for your suggestions again. Any further suggestions and feedback are appreciated.
>
> ## Intuition behind the estimators
> Thank you for your valuable suggestions. Following your suggestions, we added some intuitive explanations for each estimator and highlighted the link between them in the revised paper. These changes are marked in blue in Sections 3.2 and 3.3. It would be great if you could provide further feedback.
>
> ## Precision and Recall metrics
> Thank you for your insightful question. Comparing the generative performance under the precision and recall metrics is an interesting point. Following your guidelines, we evaluate our models in Fig. 3 under these two metrics. The results are updated in the revised paper (Appx.H.4), and are also shown in the following table.
>
> |Training schedule|Precision|Recall|
> |-|-|-|
> |EDM|0.615 |0.682|
> |DDPM|0.608	|0.683|
> |FM|0.615 |0.677|
> |SD3|0.595|0.694|
> |NCSN|0.614|0.647|
> |Ours schedule|0.626|0.667|
>
> Combining the results in Fig. 3(a,b) with the results shown in the table above, we observe that the precision metric indeed has a stronger correlation to the training loss gap in the small-noise region. Thus, our training schedule outperforms all other training schedules under this metric. In contrast, the recall metric has a stronger correlation to the training performance in the region of larger noise levels, around the critical point $\sigma^\star$, therefore the SD3 training schedule achieves the best performance. These results justify your intuition that the image quality is related to training performance at small noise scales, while recall or diversity is related to larger noise scales. We have added these discussions in our revised paper (Appx.H.4).
>
> ---
> Thank you again for your detailed suggestions and insightful questions, which help to improve the quality of our text. If you have any further questions and suggestions, we are glad to discuss with you.

---

### Meta-Review · Area_Chair_nvxC · 2026-01-06

**Summary:**

The paper addresses a fundamental ambiguity in diffusion model training—specifically that the raw loss is not indicative of data-fitting quality due to a non-zero, unknown optimal loss—by deriving a closed-form expression for this optimal value and proposing efficient estimators (DOL and cDOL). This theoretical contribution allows for the decomposition of aleatoric uncertainty from model capacity, enabling the formulation of a "loss gap" metric that facilitates more principled training schedules and scaling law analyses. Reviewers unanimously praised the work for moving beyond empirical heuristics to provide a theoretically grounded framework for diagnosing and improving diffusion models. During the rebuttal, the authors successfully addressed all concerns, providing robust evidence that their improved schedules enhance perceptual quality (validated via human preference studies and Precision/Recall metrics) without increasing memorization, solidifying the paper as a significant contribution to generative modeling. Thus, the paper is recommended for acceptance.

**Reviewer Concerns:**

Addressed Concerns: (1) Presentation and Clarity: Reviewers AWVe, LMDA, and yn5v raised issues regarding figure readability (e.g., Figure 1) and dense notation. The authors revised the manuscript, adjusted line heights, and simplified the presentation of alternative formulations. (2) Additional Metrics and Validation: Reviewers AWVe and Bp46 requested validation beyond FID. The authors included Precision/Recall analysis (showing the schedule's impact on different noise regimes) and a human preference study. (3) Memorization: Reviewer LMDA questioned if lower loss implies better generalization or just memorization. The authors provided a memorization study showing their method improves generation quality without worsening memorization. (4) Schedule Intuition: Reviewer yn5v asked for a clearer intuition behind the training schedule design. The authors clarified the distinction between the loss weight (importance) and noise schedule (optimization frequency) and provided empirical justification for the design choices.

Outstanding Concerns: None. The reviewers' concerns were predominantly addressed during the rebuttal phase. Minor formatting preferences may remain but do not impact the scientific validity.

**Reviewer Scores:**

Reviewer AWVe: 8 $\rightarrow$ 8

Reviewer LMDA: 6 $\rightarrow$ 6

Reviewer yn5v: 8 $\rightarrow$ 8

Reviewer Bp46: 6 $\rightarrow$ 6

---

### Decision · Program_Chairs · 2026-01-26

Accept (Poster)